# Globally invariant metabolism but density-diversity mismatch in springtails

Soil life supports the functioning and biodiversity of terrestrial ecosystems. Springtails (Collembola) are among the most abundant soil arthropods regulating soil fertility and flow of energy through above- and belowground food webs. However, the global distribution of springtail diversity and density, and how these relate to energy fluxes remains unknown. Here, using a global dataset representing 2470 sites, we estimate the total soil springtail biomass at 27.5 megatons carbon, which is threefold higher than wild terrestrial vertebrates, and record peak densities up to 2 million individuals per square meter in the tundra. Despite a 20-fold biomass difference between the tundra and the tropics, springtail energy use (community metabolism) remains similar across the latitudinal gradient, owing to the changes in temperature with latitude. Neither springtail density nor community metabolism is predicted by local species richness, which is high in the tropics, but comparably high in some temperate forests and even tundra. Changes in springtail activity may emerge from latitudinal gradients in temperature, predation and resource limitation in soil communities. Contrasting relationships of biomass, diversity and activity of springtail communities with temperature suggest that climate warming will alter fundamental soil biodiversity metrics in different directions, potentially restructuring terrestrial food webs and affecting soil functioning.

Soil biodiversity is an essential component of every terrestrial habitat, affecting nutrient cycling, soil fertility, and plant-soil feedbacks, among other ecosystem functions[1-3]. Soil functioning is jointly driven by multiple components of soil biota that are closely interconnected, including plants, microorganisms, micro-, meso-, and macrofauna[4,5]. Land use, human activities, and climate changes induce widespread fundamental changes in the abundance, diversity, and activity of soil biota, altering functional connections and ecosystem-level processes in the terrestrial biosphere[6]. To understand and adapt to these changes, comprehensive knowledge about the global distribution of multiple soil biota components is urgently needed[7,8].

With a growing understanding of the biogeography of microorganisms[9], micro-[10], and macrofauna[11], a critical knowledge gap is the global distribution of soil mesofauna. Springtails (Hexapoda: Collembola) are among the most abundant groups of mesofauna and soil animals from the equator to polar regions[12,13]. They are mostly

microbial feeders, but also graze on litter and are often closely associated with plant roots[14,15]. Through these trophic relationships, springtails affect the growth and dispersal of prokaryotes, fungi, and plants, thereby supporting nutrient cycling via the transformation, degradation, and stabilisation of organic matter[13,16]. Furthermore, springtails are a key food resource for soil- and surface-dwelling predators[13,14], thus occupying a central position in terrestrial food webs and supporting biodiversity at higher trophic levels.

To assess different functional facets of biological communities, metrics such as population density and biomass (reflecting carbon stocks), taxonomic and phylogenetic diversity (ensuring multifunctionality and stability), and metabolic activity (quantifying energy fluxes and thus functional influence) are commonly used[17-20]. Recent assessments have found unexpected global biodiversity hotspots in temperate regions for microorganisms (fungi and prokaryotes)[9] and macrofauna (earthworms)[11], which do not correspond to the common

✉ e-mail: anton.potapov@biologie.uni-goettingen.de

latitudinal biodiversity gradient found in aboveground organisms[21]. Functional complementarity principles[19] suggest that diverse soil communities in temperate ecosystems are able to support higher organismal densities and have a more efficient resource use (i.e., a higher total activity) than at other latitudes. However, there are no global assessments of soil animal metabolic activities. In contrast to expectations of complementarity principles, previous studies on plants[22,23] and microbes[9,24] suggest that diversity and activity (represented by respiration) do not co-vary at the global scale, probably because strong environmental constraints (e.g., temperature) limit this relationship. These discrepancies emphasize the need to investigate relationships of multiple metrics of soil animal communities. Springtails are an ideal model organism group for exploring such relationships at the global scale, due to their ubiquity, functional diversity, and high local species richness[12–14].

Current knowledge suggests that springtails are especially abundant and diverse in temperate coniferous forests, but less diverse in polar regions[20,25]. Many springtails are adapted to high and stable humidity, and sensitive to drought and temperature changes[26,27]. Consequently, springtail density and diversity are likely to decrease with future climate change, detrimentally affecting soil food webs and ecosystem functioning[28]. At the same time, springtail densities are

relatively high in urban areas and in agricultural fields[29,30], so global springtail biomass may be moderately affected by land-use changes worldwide. Disentangling the roles of vegetation, climate, human disturbance, and other predictors of various springtail community metrics will be critical to understand their contribution to soil functioning under different global change scenarios[7,10].

Here, we report global projections of density, diversity, and metabolic activity of soil springtail communities, and test whether high species richness supports increased density and total activity (i.e. community metabolism) across springtail communities globally, or whether this relationship is constrained by environmental and biotic controls. We aimed (1) to assess whether the global distribution of springtail diversity matches that of aboveground biota or other soil animals; (2) to test how different metrics of springtail communities are affected by climate and human activities; and (3) to quantify the global biomass of springtails as a component of the global carbon stock. Using an extensive dataset of soil springtail communities collected within the framework of the #GlobalCollembola initiative[13] (2470 sites and 43,601 samples across all continents; Fig. 1a), we show contrasting patterns across soil biodiversity metrics at the global scale and demonstrate that springtails are among the most functionally important and ubiquitous animals in the terrestrial biosphere.

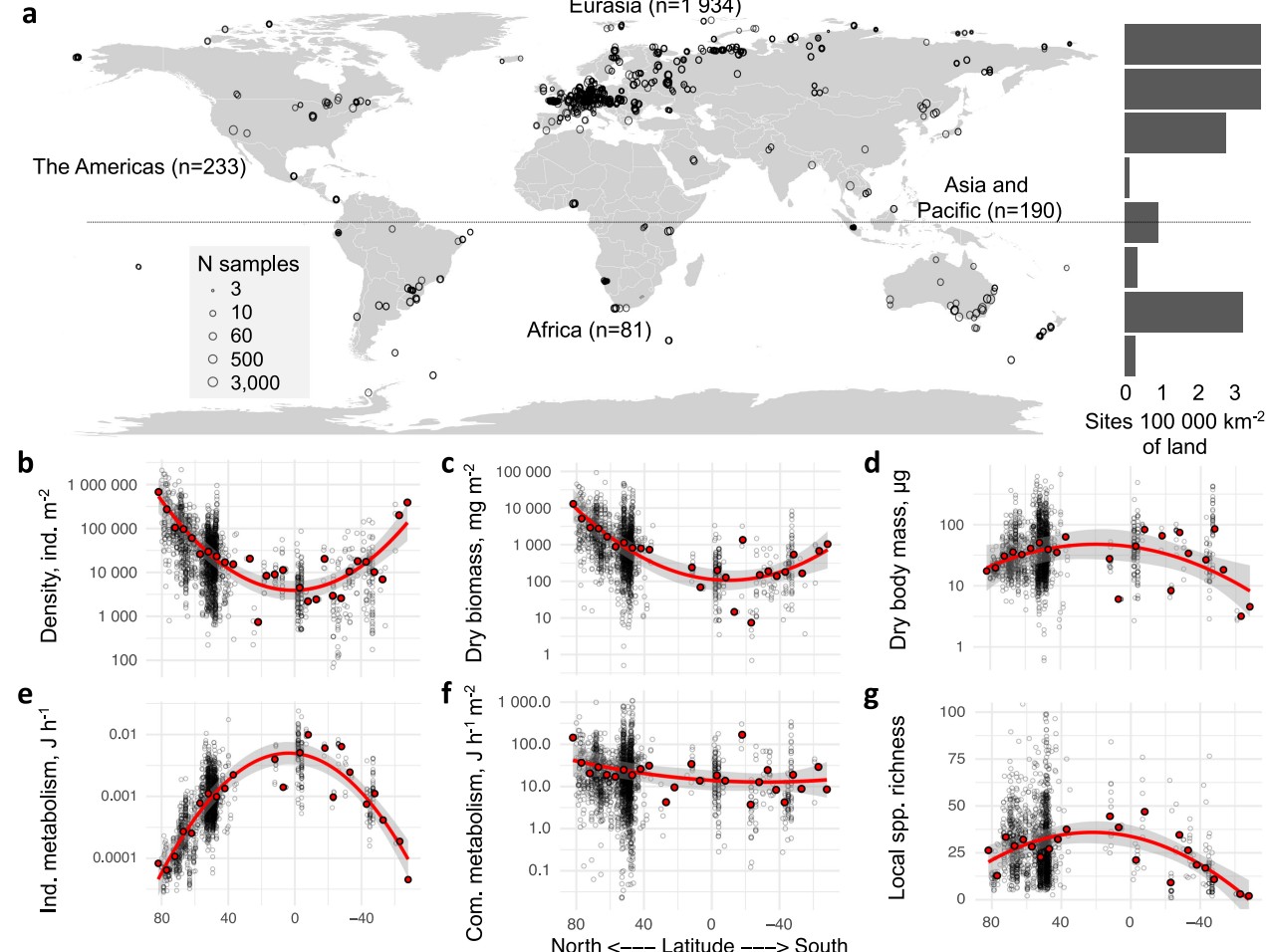

**Fig. 1 | Sampling locations and latitudinal gradients in springtail community metrics. a** Distribution of the 2470 sampling sites (43,601 soil samples). The histogram shows the number of sites in each 20-degree latitudinal belt, relative to the total land area in the belt. **b–g**, Variation in density (*n* = 2210 independent sites), biomass, community metabolism, average body mass and average individual metabolism (*n* = 2053), and local species richness (*n* = 1735) with latitude. Grey circles across panels show sampling sites; red points are averages for 5-degree latitudinal belts; trends are illustrated with a quadratic function based on 5-degree averages (red line shows the mean, shaded band shows the 95% confidence interval). Source data are provided as a Source data file.

## Results and discussion

### Latitudinal gradient

To calculate total biomass and metabolism of each springtail community, we used recorded population densities together with estimated individual body masses and metabolic rates. Body masses and metabolic rates were derived from taxon-specific body lengths using mean annual soil temperature and allometric regressions (for calculations and parameter uncertainties see Methods). For the assessment of local species richness, we selected 70% of the sampling sites with taxonomically-resolved communities and calculated rarefaction curves to account for unequal sampling efforts; we also performed analyses using raw species richness data from a subset of samples. As such, our trends refer to local diversity (hundreds of metres), but may not be representative of regional-level diversity[31].

Springtail density varied c. 30-fold across latitudes (Fig. 1b), with maximum densities in tundra (median = 131,422 individuals m$^{-2}$) and minimum densities in tropical forests (5831 individuals m$^{-2}$) and agricultural ecosystems (3438 individuals m$^{-2}$; Supplementary Fig. 2; $n$ = 2210). Springtail dry biomass followed the same trend, with c. 20-fold higher biomass in tundra (median = 3.09 g m$^{-2}$) compared to tropical agricultural and forest ecosystems (c. 0.16 g m$^{-2}$), due to a lower average community body mass in polar as opposed to temperate and tropical ecosystems (Fig. 1c, d; Supplementary Fig. 2; $n$ = 2053). These density and biomass estimates are in line with earlier reported cross-biome comparisons[20], confirming these trends across wider environmental gradients. The difference in average community body mass may be explained by lower proportion of large surface-dwelling springtail genera in polar regions[32].

Being dependent on temperature and body mass, average individual metabolism was approximately 20 times higher in tropical than in polar ecosystems (Fig. 1e), which resulted in similar community metabolism across the latitudinal gradient (Fig. 1f; total $n$ = 2053). Hence, tropical springtail communities expend a similar amount of energy per unit time and area as polar communities, despite having 20-fold lower biomass. This striking pattern resembles aboveground ecosystem respiration, which also changes little across the global air temperature gradient[23]. High metabolic rates but low densities of springtail communities are consistent with the high soil respiration rates and low litter accumulation in the tropics compared to biomes at higher latitudes[8,24]. Litter removal is facilitated by soil animals, which have to consume more food per unit biomass to meet their metabolic needs under high tropical temperatures[33] and thus enhance decomposition in wet and warm tropical ecosystems[34]. This suggests that soil animal communities in the tropics are under strong bottom-up control (by the amount and quality of litter), but also under strong top-down control by predators, which likewise have to feed more at high temperatures[33,35]. By contrast, polar communities have access to ample organic matter stocks[8], are under weaker top-down control[33,35], but their activity is constrained by the cold environment. The latitudinal gradient in environmental and biotic controls may explain why community metabolism did not increase as expected towards warm tropical ecosystems.

We found only weak latitudinal trends in local species richness (extrapolated values), which was highest in tropical forests (mean = 36.6 species site$^{-1}$) and lowest in temperate agricultural (19.5 species site$^{-1}$) and grassland ecosystems (22.8 species site$^{-1}$; Fig. 1g; Supplementary Fig. 2). Generally, the similar local diversity in different climates deviates from the latitudinal biodiversity gradients reported for aboveground and aquatic taxa[21,22], and corroborates the hypothesized mismatch between above- and belowground biodiversity distributions[36]. This mismatch calls for explicit assessments of soil biodiversity hotspots for monitoring and conservation of soil organisms[7].

### Global distribution and its predictors

To map the global distribution of springtail community metrics and uncover its predictors, we pre-selected climatic, vegetation, soil, topographic, and anthropogenic variables with known ecological effects on springtails (Supplementary Fig. 9a). To perform a global extrapolation, we used 22 of the pre-selected variables that were globally available and applied a random forest algorithm to identify the strongest spatial associations of community parameters with environmental layers[10]. To reveal the key driving factors of springtail communities, we ran a path analysis with 12 non-collinear variables (Supplementary Fig. 9b). The European spatial clustering in our data distribution (Fig. 1a), was taken in consideration with a continental-scale validation in both analyses (see Methods). In addition, we ran linear modelling on a subset of data to explore the effect of seasonal climate variation and sampling methodology.

At the global scale, species richness was not related to biomass (Pearson's $R^2$ = 0.02) or density (Pearson's $R^2$ = 0.03 for extrapolated and $R^2$ = 0.07 for raw species richness; Fig. 2a). Our extrapolations revealed at least five types of geographical areas with specific combinations of density and species richness patterns (Fig. 2a): (1) polar regions with very high densities and medium-to-high species richness such as the Arctic; (2) temperate regions with medium densities and high species richness such as mountainous and forested areas in Europe, Asia, and North America; (3) temperate regions with medium to high densities but moderate species richness such as arid temperate biomes (e.g., dry grasslands); (4) temperate, subtropical, and tropical arid ecosystems with low densities and species richness such as semi-deserts and other arid regions (largely masked on the map); (5) tropical areas with low densities but high species richness such as tropical forests and grasslands. Hotspots of springtail community metabolism were observed across a range of different latitudes (Fig. 2b), but were not associated with biodiversity hotspots (Pearson's $R^2$ < 0.01 for extrapolated and $R^2$ = 0.07 for raw species richness), emphasizing that species richness is neither associated with higher density nor metabolism of springtail communities at the global scale.

Path analysis suggested that springtail density increases with latitude, NDVI (vegetation biomass), and soil pH, but decreases with increasing mean annual air temperature, aridity index (under dryer conditions), and elevation (Fig. 3; similar responses were obtained by linear modelling; Supplementary Fig. 10). The negative global relationship of density with aridity was expected for physiologically moisture-dependent animals such as springtails[26], and was also observed in nematodes[10]. Similar to patterns for earthworms[11], soil properties had less evident linear effects on springtail density than climate at the global scale. However, the relationships of density with soil pH and organic carbon content were hump-shaped, suggesting that intermediate values of these parameters are optimal for springtails (Supplementary Fig. 8), which is also observed for nematodes[10]. Unfortunately, we could not evaluate the effects of nutrient elements such as nitrogen and phosphorus on springtail communities due to a lack of independent global assessments of these properties. Performing them would be an important step towards understanding the soil biosphere. Existing evidence points to soil properties as key predictors of microfauna (nematodes)[10], climate as a key predictor of macrofauna (earthworms)[11], and a combination of both as predictors of mesofauna (springtails) at the global scale.

Springtail density and biomass were lower in woodlands, grasslands, and agricultural sites in comparison to scrub-dominated landscapes (Fig. 3). In contrast to previous global assessments of soil animal biodiversity[10,11], tundra was extensively sampled in our dataset ($n$ = 253; Fig. 1a), and densities >1 million individuals per square metre were recorded at 12 independent sites. The high species richness of tundra communities (Fig. 2a) suggests a long evolutionary history of springtails in cold climates; indeed, they are currently the most taxonomically represented group of terrestrial arthropods in the Arctic[32]

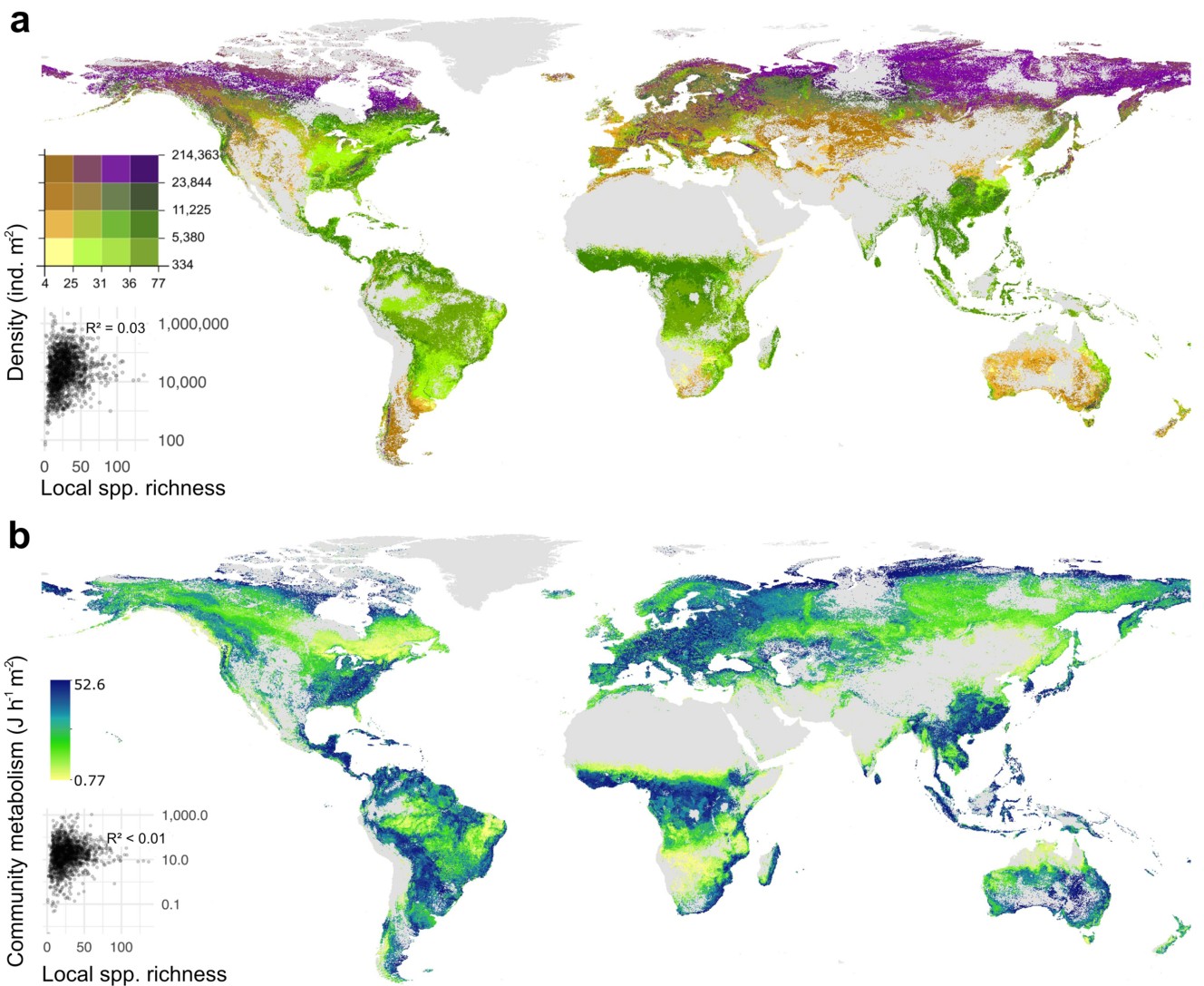

**Fig. 2 | Global maps overlapping modelled springtail density and local species richness and community metabolism in soil.** In (**a**) colours distinguish areas with different combinations of density and species richness, e.g., low density−low richness is given in yellow and high density−high richness in violet. In (**b**) the colour gradient indicates community metabolism, with potential coldspots shown in yellow and hotspots shown in blue. Pixels below the 90% extrapolation threshold for the corresponding variables are masked (see methods). Correlations between density or metabolism and species richness (inset graphs) are based on site-level data (points; $n = 1257$).

and the Antarctic[37]. Tundra remains under snow cover for most of the year, flourishing during summer when high springtail densities were recorded. During winter, springtails survive under the snow using remarkable adaptations to subzero temperatures (dehydration and supercooling[38]). Our linear modelling showed that the effect of seasonal climatic variation on springtail density and biomass is limited in comparison to the global variation in annual means (Supplementary Fig. 10), and that model with quadratic relationship with mean annual temperature explains better observed patterns in extrapolated species richness than a linear one (AIC 9611 vs 9501). However, seasonal climatic variation has critical effects on springtail activity (Supplementary Fig. 10), suggesting that functioning of the soil ecosystem is highly dynamic in time. Importantly, tundra soils contain a major proportion of the total soil organic matter and microbial biomass stored in the terrestrial biosphere[8]. As climate warming alters carbon cycling in the tundra[39], longer active periods of springtails could accelerate soil carbon release to the atmosphere in polar regions[40].

Across tropical ecosystems in the Amazon basin, equatorial Africa, and Southeast Asia, low density and biomass of springtails were recorded and extrapolated (Fig. 2a, Supplementary Figs. 4 and 6).

Mesofauna in general have low abundances in tropical ecosystems, where the litter layer is shallow and larger soil-associated invertebrates, such as earthworms, termites, and ants, commonly dominate[20]. Our study supports this trend also found in recent global assessments of other soil invertebrates[10,11,41]. However, considering the high mass-specific metabolism of springtails and high predation rates in tropical communities[18,25,33], a quantitative comparison of energy flows and stocks across latitudes and groups of soil fauna is needed.

Interestingly, we found no pronounced influence of agriculture and human population on springtail communities at the global scale; agriculture tended to have a marginally positive impact on biomass but a negative impact on species richness, although these trends were statistically significant only in some of model iterations (Fig. 3). Agricultural sites had similar springtail densities compared to woodlands and grasslands in the temperate zone (ca. 15−25k individuals m⁻²; Supplementary Fig. 3), which may be explained by large variation in management within each of these habitat types and reduced competition with more sensitive soil invertebrate groups. Some springtail species effectively survive in agricultural fields[30], where they are involved in nutrient cycling and serve as natural biocontrol agents by

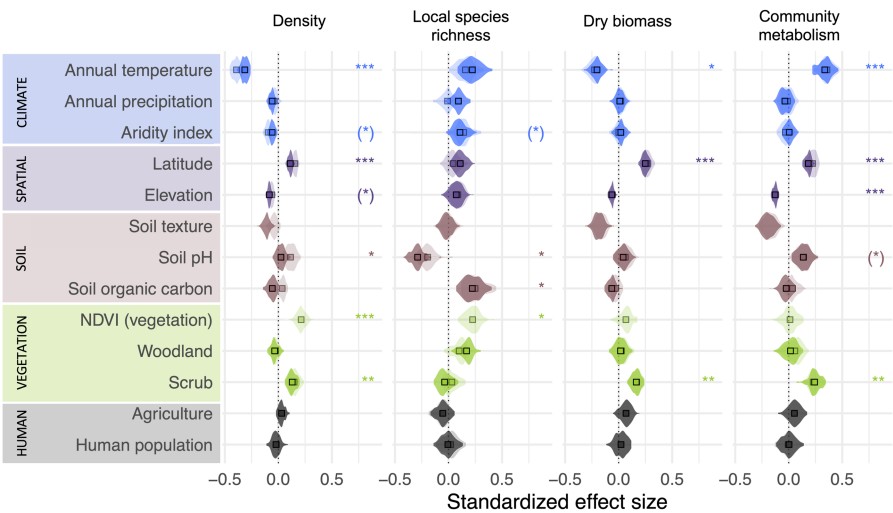

**Fig. 3 | Environmental predictors of springtail communities at the global scale.** Standardized effect sizes for direct (semi-transparent colour) and total (direct and indirect, solid colour) effects from path analysis are shown for density ($R^2 = 0.36 \pm 0.01$, $n = 723$ per iteration), local species richness ($R^2 = 0.20 \pm 0.02$, $n = 352$), biomass ($R^2 = 0.40 \pm 0.02$, $n = 568$), and community metabolism ($R^2 = 0.17 \pm 0.02$, $n = 533$). Mean values (squares) and data distribution (violins) are shown. Asterisks denote factors with a significant direct effect (two-tailed; $p < 0.05$) on a given springtail community metric for >25%(*), >50%*, >75%** and >95%*** of iterations. Source data are provided as a Source data file.

grazing on pathogenic fungi[42] and supporting arthropod predators[43]. Springtails are also commonly found in urban areas[29]. However, negative effects of agriculture and other human activities are supported by the moderate predicted local species richness in many areas of highly transformed landscapes in Europe and North America (Fig. 2). Also, our linear modelling that explicitly accounted for sampling months and methods suggested negative effects of agriculture on density and extrapolated and raw species richness of springtails (up to −40%; Supplementary Fig. 10). Overall, the negative trend in species richness at human-modified sites suggests that intensive land use may reduce springtail diversity, which is indeed often recorded[29,30,44].

The only variable that was positively associated with both density and local species richness of springtails in the path analysis was NDVI (as a proxy for vegetation biomass), reinforcing the close connection between springtail communities and the vegetation[15]. Overall, high local species richness was predicted in warm, acidic woodlands with high soil organic carbon stocks (Fig. 3). Geospatial extrapolation emphasized tropical regions and some boreal forests in North America and Eurasia as springtail diversity hotspots (Supplementary Fig. 5). In our dataset, sites with the highest extrapolated local species richness (i.e., >100 species) were located in European woodlands (Czech Republic, Slovakia). However, this picture may be biased by the historical clustering of taxonomic expertise in Europe[13]. Outside Eurasia, species-rich sites (i.e., 60–80 species) were located in Vietnamese monsoon forests and some Brazilian rainforests, but 70–90% of species in tropical communities remain undescribed[45,46]. Hence, despite low springtail density, tropical forests contribute substantially to global springtail diversity but the full extent of this contribution is unknown. Our linear modelling also demonstrated that correct estimation of density and especially species richness critically depends on the sufficient sampling area and sampling of litter and soil layers (Supplementary Fig. 10).

Our extrapolations suggest that there are c. $2 \times 10^{18}$ soil springtails globally, and their total biomass comprises c. 27.5 Mt C (16.2–28.8 Mt C minimum and maximum estimates), which corresponds to c. 190 Mt fresh weight, with respiration of c. 15.2 Mt C month⁻¹ (i.e. c. 0.2% of the global soil respiration[24]; 14.6–18.6 Mt C month⁻¹ minimum and maximum estimates). An insufficient representation of specific environmental combinations by our global extrapolation (Fig. 2) could have biased these numbers, however, most of the underrepresented areas

are covered with arid biomes where densities of springtails are very low. Our biomass estimates are very similar to the global estimated biomass of nematodes (c. 31 Mt C[10]), but lower than that of earthworms (c. 200 Mt C[11]), and exceeding by far that of all wild terrestrial vertebrates (c. 9 Mt C)[17], demonstrating that springtails are among the most abundant, biomass-rich, and ubiquitous animals on Earth.

Overall, our global dataset on soil springtail communities synthesizes the work of soil animal ecologists across the globe. It presents another milestone towards understanding the functional composition of global soil biodiversity. Being highly abundant in polar regions and some human-modified landscapes, springtails are facing two main global change frontiers: warming in the polar regions, and land-use change and urbanization in temperate and tropical regions. While the global abundance and biomass of springtails may decline with climate warming and/or vegetative biomass reduction in the coming decades, their global activity may remain unchanged. The global diversity of springtails will depend on the balance between anthropogenic transformations and conservation efforts of biomes worldwide.

## Methods
### Data reporting
The data underpinning this study is a compilation of existing datasets and therefore, no statistical methods were used to predetermine sample size, the experiments were not randomized and the investigators were not blinded to allocation during experiments and outcome assessment. The measurements were taken from distinct samples, repeated measurements from the same sites were averaged in the main analysis.

### Inclusion & ethics
Data were primarily collected from individual archives of contributing co-authors. The data collection initiative was openly announced via the mailing list of the 10th International Seminar on Apterygota and via social media (Twitter, Researchgate). In addition, colleagues from less explored regions (Africa, South America) were contacted via personal networks of the initial authors group and literature search. All direct data providers who collected and standardised the data were invited as co-authors with defined minimum role (data provision and cleaning, manuscript editing and approval). For unpublished data, people who were directly involved in sorting and identification of springtails,

including all local researchers, were invited as co-authors. Principal investigators were normally not included as co-authors, unless they contributed to conceptualisation and writing of the manuscript. All co-authors were informed and invited to contribute throughout the research process—from the study design and analysis to writing and editing. The study provided an inclusive platform for researchers around the globe to network, share and test their research ideas.

## Data acquisition

Both published and unpublished data were collected, using raw data whenever possible entered into a common template. In addition, data available from Edaphobase[47] was included. The following minimum set of variables was collected: collectors, collection method (including sampling area and depth), extraction method, identification precision and resources, collection date, latitude and longitude, vegetation type (generalized as grassland, scrub, woodland, agriculture and other for the analysis), and abundances of springtail taxa found in each soil sample (or sampling site). Underrepresented geographical areas (Africa, South America, Australia and Southeast Asia) were specifically targeted by a literature search in the Web of Science database using the keywords 'springtail' or 'Collembola', 'density' or 'abundance' or 'diversity', and the region of interest; data were acquired from all found papers if the minimum information listed above was provided. All collected datasets were cleaned using OpenRefine v3.3 (https://openrefine.org) to remove inconsistencies and typos. Geographical coordinates were checked by comparing the dataset descriptions with the geographical coordinates. In total, 363 datasets comprising 2783 sites were collected and collated into a single dataset (Supplementary Fig. 1).

## Calculation of community parameters

Community parameters were calculated at the site level. Here, we defined a site as a locality that hosts a defined springtail community, is covered by a certain vegetation type, with a certain management, and is usually represented by a sampling area of up to a hundred metres in diameter, making species co-occurrence and interactions plausible. To calculate density, numerical abundance in all samples was averaged and recalculated per square metre using the sampling area. Springtail communities were assessed predominantly during active vegetation periods (i.e., spring, summer and autumn in temperate and boreal biomes, and summer in polar biomes). Our estimations of community parameters therefore refer to the most favourable conditions (peak yearly densities). This seasonal sampling bias is likely to have little effect on our conclusions, since most springtails survive during cold periods[38,48]. Finally, we used mean annual soil temperatures[49] to estimate the seasonal mean community metabolism (described below) and tested for the seasonal bias in additional analysis (see Linear mixed-effects models).

All data analyses were conducted in R v. 4.0.2[50] with RStudio interface v. 1.4.1103 (RStudio, PBC). Data was transformed and visualised with *tidyverse* packages[51,52], unless otherwise mentioned. Background for the global maps was acquired via the *maps* package[53,54]. To calculate local species richness, we used data identified to species or morphospecies level (validated by the expert team). Since the sampling effort varied among studies, we extrapolated species richness using rarefaction curves based on individual samples with the Chao estimator[51,52] in the vegan package[53]. For some sites, sample-level data were not available in the original publications, but site-level averages were provided, and an extensive sampling effort was made. In such cases, we predicted extrapolated species richness based on the completeness (ratio of observed to extrapolated richness) recorded at sites where sample-level data were available (only sites with 5 or more samples were used for the prediction). We built a binomial model to predict completeness in sites where no sample-level data were available using latitude and the number of samples taken at a site as

predictors: *glm(Completeness~N_samples\*Latitude)*. We found a positive effect of the number of samples (Chisq = 1.97, $p = 0.0492$) and latitude (Chisq = 2.07, $p = 0.0391$) on the completeness (Supplementary Figs. 17–19). We further used this model to predict extrapolated species richness on the sites with pooled data (435 sites in Europe, 15 in Australia, 6 in South America, 4 in Asia, and 3 in Africa).

To calculate biomass, we first cross-checked all taxonomic names with the collembola.org checklist[55] using fuzzy matching algorithms (*fuzzyjoin* R package[56]) to align taxonomic names and correct typos. Then we merged taxonomic names with a dataset on body lengths compiled from the BETSI database[57], a personal database of Matty P. Berg, and additional expert contributions. We used average body lengths for the genus level (body size data on 432 genera) since data at the species level were not available for many morphospecies (especially in tropical regions), and species within most springtail genera had similar body size ranges. Data with no genus-level identifications were excluded from the analysis. Dry and fresh body masses were calculated from body length using a set of group-specific length-mass regressions (Supplementary Table 1)[58,59] and the results of different regressions applied to the same morphogroup were averaged. Dry mass was recalculated to fresh mass using corresponding group-specific coefficients[58]. We used fresh mass to calculate individual metabolic rates[60] and account for the mean annual topsoil (0–5 cm) temperature at a given site[61]. Group-specific metabolic coefficients for insects (including springtails) were used for the calculation: normalization factor (i0) ln(21.972) [J h$^{-1}$], allometric exponent (a) 0.759, and activation energy (E) 0.657 [eV][60]. Community-weighted (specimen-based) mean individual dry masses and metabolic rates were calculated for each sample and then averaged by site after excluding 10% of maximum and 10% of minimum values to reduce impact of outliers. To calculate site-level biomass and community metabolism, we summed masses or metabolic rates of individuals, averaged them across samples, and recalculated them per unit area (m$^2$).

## Parameter uncertainties

Our biomass and community metabolism approximations contain several assumptions. To account for the uncertainty in the length-mass and mass-metabolism regression coefficients, in addition to the average coefficients, we also used maximum (average + standard error) and minimum coefficients (average−standard error; Supplementary Table 1) in all equations to calculate maximum and minimum estimations of biomass and community metabolism reported in the main text. Further, we ignored latitudinal variation in body sizes within taxonomic groups[62]. Nevertheless, latitudinal differences in springtail density (30-fold), environmental temperature (from −16.0 to +27.6 °C in the air and from −10.2 to +30.4 °C in the soil), and genus-level community compositions (there are only few common genera among polar regions and the tropics)[55] are higher than the uncertainties introduced by indirect parameter estimations, which allowed us to detect global trends. Although most springtails are concentrated in the litter and uppermost soil layers[20], their vertical distribution depends on the particular ecosystem[63]. Since sampling methods are usually ecosystem-specific (i.e. sampling is done deeper in soils with developed organic layers), we treated the methods used by the original data collectors as representative of a given ecosystem. Under this assumption, we might have underestimated the number of springtails in soils with deep organic horizons, so our global estimates are conservative and we would expect true global density and biomass to be slightly higher. To minimize these effects, we excluded sites where the estimations were likely to be unreliable (see data selection below).

## Data selection

Only data collection methods allowing for area-based recalculation (e.g. Tullgren or Berlese funnels) were used for analysis. Data from artificial habitats, coastal ecosystems, caves, canopies, snow surfaces,

and strong experimental manipulations beyond the bounds of naturally occurring conditions were excluded (Supplementary Fig. 1). To ensure data quality, we performed a two-step quality check: technical selection and expert evaluation. Collected data varied according to collection protocols, such as sampling depth and the microhabitats (layers) considered. To technically exclude unreliable density estimations, we explored data with a number of diagnostic graphs (Supplementary Table 2; Supplementary Figs. 12–20) and filtered it, excluding the following: (1) All woodlands where only soil or only litter was considered; (2) All scrub ecosystems where only ground cover (litter or mosses) was considered; (3) Agricultural sites in temperate zones where only soil with sampling depth <10 cm was considered. Additionally, 10% of the lowest values were individually checked and excluded if density was unrealistically low for the given ecosystem (outliers with density over three times lower than 1% percentile within each ecosystem type). In total, 237 sites were excluded from density, and 394 sites from biomass, and community metabolism analyses based on these criteria (Supplementary Figs. 15 and 16). For the local species richness estimates, we removed all extrapolations based on sites with fewer than three samples and no (morpho)species identifications (647 sites; Supplementary Figs. 1 and 20).

## Data expert evaluation

We performed manual expert evaluation of every contributed dataset. Evaluation was done by an expert board of springtail specialists, each with extensive research experience in a certain geographic area: Anatoly Babenko−high latitude regions in both north and south hemispheres; Bruno Bellini−Central and South America; Jean-François Ponge−Central and Western Europe; Louis Deharveng−Africa and Asia; Lubomir Kovac−Southern Europe; Mikhail Potapov and Natalia Kuznetsova−Eastern and Northern Europe. Each dataset was scored by the experts separately for density and species richness estimation as either trustworthy, acceptable, or unreliable. Density estimation quality was assessed using information about the sampling and extraction method and the density estimation itself. Species richness estimation quality was assessed using information about the identification key, experience of the person who identified the material, species (taxa) list, and the species richness estimation itself. Based on the expert opinions, unreliable estimates of density (together with biomass and community metabolism) and species richness were excluded (Supplementary Fig. 1). The resulting final dataset included 2470 sites and 43,601 samples[64] with a median of six samples collected at each site. The dataset comprised 2210 sites with density estimation (69–2,181,600 individuals m$^{-2}$), 2,053 sites with mean fresh body mass (1.8–3110 µg), mean metabolic rate (0.028–2.4 mJ h$^{-1}$), dry biomass (0.5–93,000 mg m$^{-2}$), fresh biomass (1.6–277,000 mg m$^{-2}$) and community metabolism estimations (0.03–1000 J h$^{-1}$), and 1735 sites with local species richness estimation (1–136.7 species; Supplementary Figs. 1 and 2). The dataset covered all major biomes (Supplementary Fig. 3), years 1970–2019, and all months: 8% of the samples were taken between December and February, 14% between March and May, 55% between June and August, and 23% between September and November (see Data availability).

## Data transformation

All parameters except for extrapolated local species richness were highly skewed (e.g., density had a global median of 21,016 individuals m$^{-2}$ and a mean of 60,454 individuals m$^{-2}$) and we applied $\log_{10}$-transformation prior to analysis. This greatly improved the fit of all statistical analyses.

## Latitudinal and ecosystem trends

To explore changes in springtail communities with latitude, we sliced the global latitudinal gradient into 5-degree bins and calculated average parameters across sites in each bin after trimming to ensure the

same statistical weight for each latitudinal bin while plotting the gradient. The latitudinal gradient was plotted with *ggplot2*[65], and quadratic smoothers were used to illustrate trends. Mean parameters of springtail communities were compared across ecosystem types using a linear model and multiple comparisons with the Tukey HSD test using *HSD.test* in the *agricolae* package[66]. Habitats were classified according to the vegetation types. Climates were classified as polar (beyond the polar circles, i.e., more than 66.5 and less than −66.5 degrees), temperate (from the polar circles to the tropics of Capricorn/Cancer, i.e. to 23.5 and −23.5 degrees) and tropical (in between 23.5 and −23.5 degrees). Habitats and climates were combined to produce ecosystem types. For the analysis, only well-represented ecosystem types were retained: polar scrub ($n = 253$), polar grassland ($n = 39$), polar woodland ($n = 28$), temperate woodland ($n = 907$), temperate scrub ($n = 104$), temperate grassland ($n = 445$), temperate agriculture ($n = 374$), tropical agriculture ($n = 68$) and tropical forest ($n = 141$; Supplementary Fig. 3).

## Selection of environmental predictors

To assess the predictors of global distributions of springtail community metrics, we pre-selected variables with a known ecological effect on springtail communities (based on expert opinions) and constructed a hypothetical relationship diagram (Supplementary Fig. 9a). Environmental data were very heterogeneous across the springtail studies, so we used globally available climatic and other environmental layers. Overall, we included global layers bearing the following information: climate (mean annual air temperature, air temperature seasonality, air temperature annual range, mean annual precipitation, precipitation seasonality, precipitation of the driest quarter[67], inversed aridity index[68]), topography (elevation, roughness[69]), vegetation and land cover (aboveground biomass[70], tree cover[71], Net Primary Production, Normalized Difference Vegetation Index [NDVI][72]), topsoil physico-chemical properties (0–15 cm depth C to N ratio, pH, clay, sand, coarse fragments, organic carbon, bulk density[73]) and human population density[74]. Some of environmental layers could not be included due to the lack of appropriate data. For example, soil phosphorus and nitrogen concentrations had to be omitted. While the global distribution of soil nitrogen concentration is available[73], it is a modelled product, which strongly correlates with soil carbon concentration, and thus cannot be used as an independent predictor.

## Geospatial global projections

To create global spatial predictions of springtail density, species richness, biomass, and community metabolism, we followed the approach previously used for nematodes[10,75] that is based on spatial associations of community parameters with global environmental information. The analysis for geospatial modelling was done in Python version 3.6.5 (Python software foundation). A Random Forest algorithm was applied to identify the spatial associations and extrapolate local observations to the global scale at the 30 arcsec resolution (approximately 1 km$^2$ pixels)[18,75]. After retrieving the environmental variable values for each location, we trained 18 model versions, each with different hyperparameter settings, i.e., variables per split (range: 2–7); minimum leaf population (range: 3–5). To minimize the potential bias of a single model, we used an ensemble of the top 10 best-performing models, selected based on the coefficient of determination (R$^2$), to create global predictions of each of the community parameters.

Model performance was assessed by 10-fold cross validation, with folds assigned randomly. The R$^2$ values for each of the five response variables were in the range of 0.30–0.57 (density: 0.567, dry biomass: 0.463, community metabolism: 0.359, extrapolated species richness: 0.302). For some of the modelled variables we observed positive spatial autocorrelation: at ranges below 150 km for density, below 100 km for community metabolism and below 150 km for extrapolated

species richness (Supplementary Note). Yet, the Moran's I values were very close to zero (the highest value was 0.07), indicating that the effect of spatial autocorrelation was very weak. These results were obtained by performing Moran's I tests using the *spatialRF* package in R[76]. To investigate the effect of spatial autocorrelation on model performance we performed a buffered leave-one-out cross-validation tests (described in detail as an alternative performance statistic for models with potential spatial autocorrelation[77,78]). As expected, the predictive power declined with increasing buffer sizes. At the scales at which we observe positive autocorrelation, i.e., where we have significant Moran's I values, coefficient of determination remained positive.

To reduce potential artifacts produced by extrapolation, geographical regions with climatic conditions poorly represented by our sites and without NPP data were excluded from the extrapolation (e.g., Sahara, Arabian desert, Himalayas). We evaluated our extrapolation quality based on spatial approximations of interpolation versus extrapolation[75]. In this approach, we first determined the range of environmental conditions represented by the observations. Next, we classified all pixels to fall within or outside the training space, in univariate and multivariate space. For the latter, we first transformed the data into principal component space, and selected the first 11 PC axes, collectively explaining 90% of the variation. Finally, we classified pixels to fall within or outside the convex hulls drawn around each possible bivariate combination of these 11 PC axes; pixels that fell outside the convex hulls in >90% of cases were masked on the main maps; for the map with density-species richness visualisation, two corresponding masks were applied (Fig. 2).

To estimate spatial variability of our predictions while accounting for the spatial sampling bias in our data (Fig. 1a) we performed a spatially stratified bootstrapping procedure. We used the relative area of each IPBES[79] region (i.e., Europe and Central Asia, Asia and the Pacific, Africa, and the Americas) to resample the original dataset, creating 100 bootstrap resamples. Each of these resamples was used to create a global map, which was then reduced to create mean, standard deviation, 95% confidence interval, and coefficient of variation maps (Supplementary Figs. 4–7).

Global biomass, abundance, and community metabolism of springtails were estimated by summing predicted values for each 30 arcsec pixel[10]. Global community metabolism was recalculated from joule to mass carbon by assuming 1 kg fresh mass = $7 \times 10^6$ J[80], an average water proportion in springtails of 70%[58], and an average carbon concentration of 45% (calculated from 225 measurements across temperate forest ecosystems)[81]. We repeated the procedure of global extrapolation and prediction for biomass and community metabolism using minimum and maximum estimates of these parameters from regression coefficient uncertainties (see Parameter uncertainties).

### Path analysis
To reveal the predictors of springtail communities at the global scale, we performed a path analysis. After filtering the selected environmental variables (see above) according to their global availability and collinearity, 13 variables were used (Supplementary Fig. 9b): mean annual air temperature, mean annual precipitation (CHELSA database[67]), aridity (CGIAR database[68]), soil pH, sand and clay contents combined (sand and clay contents were co-linear in our dataset), soil organic carbon content (SoilGrids database[73]), NDVI (MODIS database[72]), human population density (GPWv4 database[74]), latitude, elevation[69], and vegetation cover reported by the data providers following the habitat classification of European Environment Agency (woodland, scrub, agriculture, and grasslands; the latter were coded as the combination of woodland, scrub, and agriculture absent). Before running the analysis, we performed the Rosner's generalized extreme Studentized deviate test in the

*EnvStats* package[82] to exclude extreme outliers and we z-standardized all variables (Supplementary R Code).

Separate structural equation models were run to predict density, dry biomass, community metabolism, and local species richness in the *lavaan* package[83]. To account for the spatial clustering of our data in Europe, instead of running a model for the entire dataset, we divided the data by the IPBES[79] geographical regions and selected a random subset of sites for Eurasia, such that only twice the number of sites were included in the model as the second-most represented region. We ran the path analysis 99 times for each community parameter with different Eurasian subsets (density had $n = 723$ per iteration, local species richness had $n = 352$, dry biomass had $n = 568$, and community metabolism had $n = 533$). We decided to keep the share of the Eurasian dataset larger than other regions to increase the number of sites per iteration and validity of the models. The Eurasian dataset also had the best data quality among all regions and a substantial reduction in datasets from Eurasia would result in a low weight for high-quality data. We additionally ran a set of models in which the Eurasian dataset was represented by the same number of sites as the second-most represented region, which yielded similar effect directions for all factors, but slightly higher variations and fewer consistently significant effects. In the paper, only the first version of analysis is presented. To illustrate the results, we averaged effect sizes for the paths across all iterations and presented the distribution of these effect sizes using mirrored Kernel density estimation (violin) plots. We marked and discussed effects that were significant at $p < 0.05$ in more than a given number of iterations (arbitrary thresholds were set to 25%, 50%, 75% and 95% of iterations; Fig. 3).

### Linear mixed-effects modelling
To test if our results are biased by seasonal effects, sampling methodology, and/or species richness extrapolation, we selected a subset of sampling events with known sampling year and month (2997 sampling events representing 1703 sites) and ran linear mixed-effects models for springtail density, species richness (both raw and extrapolated), biomass, and community metabolism. The models were run using the *lme4* package[84]. Data were transformed as described above and analysed using Gaussian distributions except for raw species richness, which was analysed using generalised models with Poisson distribution. Sampling site was included as random effect to account for the dependence of the sampling events coming from the same sites. We included mean monthly air temperatures (offset from the annual mean) and the sum of monthly precipitation at the sampling month as additional climatic predictors. We also included total collection area and the presence of litter (or any other soil cover such as mosses) and soil in the sample to account for methodological biases. All models were run using the full dataset ($n = 2884$ sampling events for density, $n = 2540$ for raw species richness; $n = 1708$ for extrapolated species richness; $n = 2462$ for dry biomass; $n = 2289$ for community metabolism). To test if the effect of temperature on species richness is non-linear, we additionally ran the same model including quadratic function *poly(MAT, 2)*.

### Reporting summary
Further information on research design is available in the Nature Portfolio Reporting Summary linked to this article.

## Data availability
The data that support the findings of this study have been deposited in the Figshare database[64] under CC-BY 4.0 license and accession code: https://doi.org/10.6084/m9.figshare.16850419; high-resolution maps[85] can be accessed at https://doi.org/10.6084/m9.figshare.16850446. Source data are provided with this paper.

## Code availability

Programming code for the path analysis and the geospatial modelling is available under CC-BY 4.0 from Figshare[64]: https://doi.org/10.6084/m9.figshare.16850419.

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

## Acknowledgements

The article is an outcome of the #GlobalCollembola community initiative that is voluntarily supported by researchers around the world. Data collection and analysis was supported by the Russian Science Foundation (19-74-00154 to A.P.) and by Deutsche Forschungsgemeinschaft (493345801 to A.P. and 192626868—SFB 990 to S.S.). We acknowledge support by the Open Access Publication Funds of the Göttingen University. The following funding bodies provided support for individual contributors: ARC SRIEAS Grant SR200100005 Securing Antarctica's Environmental Future to S.L.C., Slovak Scientific Grant Agency VEGA 1/0438/22 to Ľ.K., RFBR 19-516-60002 to N.A.K., Carl Tryggers Stiftelse för Vetenskaplig Forskning and Qatar Petroleum to J.M.A., BIO 27 (2013-2014)-MAGyP and PICTO 2084 (2012)-ANPCyT to V.B., DAAD-19-10 and MSM200962001 to T.C., grant TE, PN-III-P1-1.1-TE-2019-0358 to C.F., NWO grant 821.01.015 to O.F., National Natural Sciences Foundation of China No 41471037 and 41871042 to M.G., BIO 27 (2013-2014), MAGyP; PICT 2084 (2012), FONCyT to D.F.G., NRF South African National Antarctic Programme grant 110734 to M.G., Natural Resources Canada (NRCan), EcoEnergy Innovation Initiative under the Office of Energy Research and Development, and the Natural Sciences and Engineering Research Council of Canada (NSERC) to I.T.H., L.A.V. and L.R., Independent Research Fund Denmark grant no. DFF-4002-00384 to M.H., Estonian Science Foundation G9145 to M.I., SA-France bilateral grant to C.J., SA (NRF)/Russia (RFBR) Joint Science and Technology Research Collaboration project no. 19-516-60002 (FRBR) and no.

118904 (NRF) to M.P. and C.J., European Research Council (ERC), European Union's Horizon 2020 research and innovation programme (grant agreement no. 677232; to N.E.); iDiv, German Research Foundation (DFG–FZT 118, 202548816) to M.J. and N.E., French National Agency of Research (ANR) (JASSUR research project; ANR-12-VBDU-0011), «Ministère de l'Agriculture et de la Pêche» and «Ministère de l'Education Nationale de la Recherche et de la Technologie» (ACTA programme), «Ministère de l'Aménagement du Territoire et de l'Environnement» (Pnetox programme), EU-funded project, ECOGEN QLK5-CT-2002-01666 (www.ecogen.dk), "Agence de l'Environnement et de la Maîtrise de l'Énergie" (BIOINDICATEUR 2, BIOTECHNOSOL), ANDRA and GISFI (www.gisfi.fr) to S.J., GRR SER-BIODIV (Région Normandie, France) to MCha., ESF9258, B02 to A.K., Fundamental Research Funds for the Central Universities (grant no. 2018CDXYCH0014) to D.L., DFG 316045089 to J.L., Massey University Research Fund grant to M.A.M., DFG SCHE 376/38-2 to M.M.P., grant from the Austria Academy of Science: Heritage_2020-043_Modeling-Museum to P.Q., Slovak Scientific Grant Agency: VEGA Nos. 1/0441/03 and 1/3267/06 to N.R., Higher Education Commission of Pakistan to M.I.R., RSF 21-74-00126 to R.A.S., Austrian Federal Government and European Union (Rural Development 2014-2020) to J.S., AAAA-A17-122040600025-2 to A.A.T., Brazilian Council for Scientific and Technological Development—CNPq (grant no. 152717/2016-1) to B.R.W., 309030/2018-8 to D.Z. and 305426/2018-4 to B.C.B., National Natural Science Foundation of China (31970434, 31772491) to N.N.G., Research and Innovation Support Foundation of Santa Catarina (FAPESC) (6.309/2011-6/FAPESC) and the CNPq (563251/2010-7/CNPq) to L.C.I.O.F., O.K.-F., the Latvian Council of Science Grants no. 90.108, 93.140, 96.0110, 01.0344 to E.J., CNPq for the Research Productivity Grant (305939/2018-1) to D.B., FPI-MICINN grant in the project INTERCAPA (CGL2014-56739-R) to P.H, the Natural Sciences and Engineering Research Council of Canada (NSERC) to Z.L., Ministry of Innovation and Technology of Hungary TKP2021-NKTA-43 to D.W. Authors are grateful to Penelope Greenslade for providing the literature on Australian Collembola communities. Authors are grateful to Frans Janssens for providing the global checklist of Collembola.

## Author contributions

A.M.P. designed the study, coordinated the collection, cleaning and standardization of data and wrote the first draft of the manuscript. C.G. and A.M.P. designed and performed the path analysis. J.v.d.H. designed and performed the geospatial modelling. A.B., B.C.B., L.D., Ľ.K., N.A.K., J.F.P. and M.B.Pot. evaluated the data quality. M.P.B., S.L.C., J.F.P., D.J.R., T.C., N.E., S.S., M.Cha., J.F. and I.T.H. contributed to writing and conceptualisation of the manuscript. A.M.P., A.B., M.P.B., S.L.C., L.D., Ľ.K., N.A.K., J.F.P., M.B.Pot., D.J.R., J.M.A., J.I.A., I.B., V.B., S.B., T.B., G.C., M.Cha., T.Che., M.C., A.T.C., J.C., P.Č., A.M.d.l.P, D.A., S.D.F., C.F., J.F., O.F., S.F., E.G., M.G., B.G., D.F.G., M.Gre., I.T.H., C.H., M.H., P.H., M.I., C.J., M.J., S.J., B.S.J., E.J., O.F., L.C.I.O.F., O.K.-F., D.B., E.J.K., A.K., E.A.L., D.L., J.L., M.J.L., M.T.M., M.M.M., M.A.M., T.N., I.N., R.O., J.G.P., M.M.P., P.Q., N.R., M.I.R., L.J.R., L.R., R.A.S., S.Sal., E.J.S., N.S., C.S., J.S., Y.B.S., S.K.S., M.S., X.S., W.I.S., A.A.T., M.P.T., M.A.T., M.S.T., Z.L., M.N.T., A.V.U., L.A.V., L.A.W., B.R.W., D.W., D.Wu., Z.J.X., R.Y., D.Z., N.E. and S.S. contributed data. All authors contributed to editing of the paper.

## Funding

## Competing interests

The authors declare no competing interests.

## Additional information

**Anton M. Potapov** [1,2,3,4] ✉, **Carlos A. Guerra** [3,4], **Johan van den Hoogen**[5], **Anatoly Babenko** [2], **Bruno C. Bellini** [6], **Matty P. Berg**[7,8], **Steven L. Chown** [9], **Louis Deharveng**[10], **Ľubomír Kováč** [11], **Natalia A. Kuznetsova**[12], **Jean-François Ponge**[13], **Mikhail B. Potapov**[12], **David J. Russell**[14], **Douglas Alexandre**[15], **Juha M. Alatalo**[16], **Javier I. Arbea** [17], **Ipsa Bandyopadhyaya**[18], **Verónica Bernava**[19], **Stef Bokhorst** [7], **Thomas Bolger**[20,21], **Gabriela Castaño-Meneses** [22], **Matthieu Chauvat** [23], **Ting-Wen Chen**[24,1], **Mathilde Chomel**[25], **Aimee T. Classen** [26], **Jerome Cortet**[27], **Peter Čuchta**[24], **Ana Manuela de la Pedrosa**[28], **Susana S. D. Ferreira**[7], **Cristina Fiera**[29], **Juliane Filser** [30], **Oscar Franken** [7,8,31], **Saori Fujii**[32], **Essivi Gagnon Koudji**[33], **Meixiang Gao**[34], **Benoit Gendreau-Berthiaume**[35], **Diego F. Gomez-Pamies**[36], **Michelle Greve** [37], **I. Tanya Handa**[33], **Charlène Heiniger**[38], **Martin Holmstrup** [39], **Pablo Homet**[40], **Mari Ivask**[41], **Charlene Janion-Scheepers** [42,43], **Malte Jochum** [3,4], **Sophie Joimel**[44], **Bruna Claudia S. Jorge**[45], **Edite Jucevica** [46], **Olga Ferlian** [3,4], **Luís Carlos Iuñes de Oliveira Filho** [47], **Osmar Klauberg-Filho**[47], **Dilmar Baretta**[48], **Eveline J. Krab** [49,50],

Annely Kuu[51], Estevam C. A. de Lima [52], Dunmei Lin[53], Zoe Lindo [54], Amy Liu[9], Jing-Zhong Lu [1], María José Luciañez[28], Michael T. Marx[55], Matthew A. McCary[56], Maria A. Minor[57], Taizo Nakamori[58], Ilaria Negri [59], Raúl Ochoa-Hueso [60,61], José G. Palacios-Vargas [62], Melanie M. Pollierer [1], Pascal Querner[63,64], Natália Raschmanová[11], Muhammad Imtiaz Rashid[65], Laura J. Raymond-Léonard [33], Laurent Rousseau[33], Ruslan A. Saifutdinov[2], Sandrine Salmon[66], Emma J. Sayer [67,68], Nicole Scheunemann[1,69], Cornelia Scholz[64], Julia Seeber[70,71], Yulia B. Shveenkova[72], Sophya K. Stebaeva[2], Maria Sterzynska [73], Xin Sun[74], Winda I. Susanti[1], Anastasia A. Taskaeva [75], Madhav P. Thakur [76], Maria A. Tsiafouli [77], Matthew S. Turnbull[78], Mthokozisi N. Twala [37], Alexei V. Uvarov[2], Lisa A. Venier[79], Lina A. Widenfalk[80,81], Bruna R. Winck[45], Daniel Winkler [82], Donghui Wu[83,84,85], Zhijing Xie[83], Rui Yin [86], Douglas Zeppelini[87], Thomas W. Crowther [5], Nico Eisenhauer [3,4] & Stefan Scheu [1,88]

[1]Johann Friedrich Blumenbach Institute of Zoology and Anthropology, University of Göttingen, Göttingen, Germany. [2]A.N. Severtsov Institute of Ecology and Evolution, Russian Academy of Sciences, Moscow, Russia. [3]German Centre for Integrative Biodiversity Research (iDiv) Halle-Jena-Leipzig, Leipzig, Germany. [4]Institute of Biology, Leipzig University, Leipzig, Germany. [5]Department of Environmental Systems Science, Institute of Integrative Biology, ETH Zürich, Zürich, Switzerland. [6]Department of Botany and Zoology, Federal University of Rio Grande do Norte, Natal, RN, Brazil. [7]Department of Ecological Science, Vrije Universiteit Amsterdam, Amsterdam, the Netherlands. [8]Community and Conservation Ecology Group, Groningen Institute of Evolutionary Life Science, University of Groningen, Amsterdam, the Netherlands. [9]Securing Antarctica's Environmental Future, School of Biological Sciences, Monash University, Melbourne, Australia. [10]ISYEB, Muséum National d'Histoire Naturelle, Paris, France. [11]Department of Zoology, Institute of Biology and Ecology, Faculty of Science, Pavol Jozef Šafárik University in Košice, Košice, Slovakia. [12]Institute of Biology and Chemistry, Moscow Pedagogical State University, Moscow, Russia. [13]Département Adaptations du Vivant, Muséum National d'Histoire Naturelle, Brunoy, France. [14]Department of Soil Zoology, Senckenberg Society for Nature Research, Görlitz, Germany. [15]Department of Soil Science, Centre for Agriculture and Veterinary Science, Santa Catarina State University University (UDUESC-Lages), Lages, SC, Brazil. [16]Environmental Science Center, Qatar University, Doha, Qatar. [17]Department of Sciences, CEPA Camargo, Astillero, Spain. [18]Visva Bharati University, Bengal, India. [19]Administración de Parques Nacionales, San Antonio, Argentina. [20]School of Biology and Environmental Science, University College Dublin, Dublin, Ireland. [21]Earth Institute, University College Dublin, Dublin, Ireland. [22]Unidad Multidisciplinaria de Docencia e Investigación, Facultad de Ciencias, Campus Juriquilla, Universidad Nacional Autónoma de México, Querétaro, México. [23]Normandie University—UNIROUEN, INRAE, ECODIV, Rouen, France. [24]Biology Centre of the Czech Academy of Sciences, Institute of Soil Biology, České Budějovice, Czech Republic. [25]FiBL France, Research Institute of Organic Agriculture, Eurre, France. [26]Department of Ecology & Evolutionary Biology, University of Michigan, Ann Arbor, MI 48109, USA. [27]Centre d'Ecologie Fonctionnelle et Evolutive, Université Paul-Valéry Montpellier 3, Montpellier, France. [28]Departmento de Biología Zoología, Universidad Autónoma de Madrid, Madrid, Spain. [29]Institute of Biology Bucharest, Romanian Academy, Bucharest, Romania. [30]FB 02, UFT, General and Theoretical Ecology, University of Bremen, Bremen, Germany. [31]Department of Coastal Systems, Royal Netherlands Institute for Sea Research, 't Horntje, the Netherlands. [32]Department of Forest Entomology, Forestry and Forest Products Research Institute, Tsukuba, Japan. [33]Département des Sciences Biologiques, Université du Québec à Montréal, Québec, Canada. [34]Department of Geography and Spatial Information Techniques, Ningbo University, Ningbo, China. [35]Département des Sciences Naturelles, Université du Québec en Outaouais, Québec, Canada. [36]Instituto de Biología Subtropical, Consejo Nacional de Investigaciones Científicas y Técnicas-Universidad Nacional de Misiones, Puerto Iguazú, Argentina. [37]Department of Plant and Soil Sciences, University of Pretoria, Pretoria, South Africa. [38]HES-SO University of Applied Sciences and Arts Western Switzerland, Geneva, Switzerland. [39]Section of Terrestrial Ecology, Department of Ecoscience, Aarhus University, Aarhus, Denmark. [40]Instituto de Recursos Naturales y Agrobiología de Sevilla (IRNAS), Consejo Superior de Investigaciones Científicas (CSIC), Sevilla, Spain. [41]Tartu College, Tallinn University of Technology, Tartu, Estonia. [42]Department of Biological Sciences, University of Cape Town, Rondebosch, South Africa. [43]Department of Entomology, Iziko Museums of South Africa, Cape Town, South Africa. [44]Université Paris-Saclay, INRAE, AgroParisTech, UMR EcoSys, Thiverval-Grignon, France. [45]Quantitative Ecology Lab, Department of Ecology, Universidade Federal do Rio Grande do Sul, Porto Alegre, Brazil. [46]Institute of Biology, University of Latvia, Riga, Latvia. [47]Department of Soil Science, Centre for Agriculture and Veterinary Science, Santa Catarina State University (UDESC-Lages), Lages, SC, Brazil. [48]Department of Animal Science, Santa Catarina State University (UDESC Oeste), Chapecó, SC, Brazil. [49]Department of Soil and Environment, Swedish University of Agricultural Sciences, Uppsala, Sweden. [50]Climate Impacts Research Centre, Department of Ecology and Environmental Science, Umeå University, Abisko, Sweden. [51]Institute of Agricultural and Environmental Sciences, Chair of Soil Science, Estonian University of Life Sciences, Tartu, Estonia. [52]Departamento de Entomologia, Museu Nacional, Universidade Federal do Rio de Janeiro, Rio de Janeiro, Brazil. [53]Key Laboratory of the Three Gorges Reservoir Region's Eco-Environment, Ministry of Education, Chongqing University, Chongqing, China. [54]Department of Biology, University of Western Ontario, London, Ontario N6A 5B7, Canada. [55]Institute of Zoology, Johannes Gutenberg University Mainz, Mainz, Germany. [56]Department of BioSciences, Rice University, Houston, USA. [57]Wildlife & Ecology Group, School of Agriculture and Environment, Massey University, Palmerston North, New Zealand. [58]Graduate School of Environment and Information Sciences, Yokohama National University, Yokohama, Japan. [59]Department of Sustainable Crop Production (DI.PRO.VE.S.), Università Cattolica del Sacro Cuore, Piacenza, Italy. [60]Department of Biology, IVAGRO, University of Cádiz, Puerto Real, Spain. [61]Department of Terrestrial Ecology, Netherlands Institute of Ecology (NIOO KNAW), Wageningen NL-6700 AB, the Netherlands. [62]Lab. Ecología y Sistemática de Microartrópodos, Depto. Ecología y Recursos Naturales, Facultad de Ciencias, Universidad Nacional Autónoma de México, México, México. [63]Natural History Museum Vienna, 1. Zoology, Vienna, Austria. [64]Department of Integrated Biology and Biodiversity Research, University of Natural Resources and Life Sciences, Vienna, Austria. [65]Center of Excellence in Environmental Studies, King Abdulaziz University, Jeddah, Saudi Arabia. [66]UMR 7179 MECADEV—AVIV department, Muséum National d'Histoire Naturelle, Brunoy, France. [67]Lancaster Environment Centre, Lancaster University, Lancaster, UK. [68]Smithsonian Tropical Research Institute, Balboa, Ancon, Panama, Republic of Panama. [69]Department of Soil Zoology, Senckenberg Museum of Natural History Görlitz, Görlitz, Germany. [70]Institute for Alpine Environment, Eurac Research, Bozen, Italy. [71]Department of Ecology, University of Innsbruck, Innsbruck, Austria. [72]State Nature Reserve "Privolzhskaya Lesostep", Penza, Russia. [73]Department of Systematics, Zoogeography and Ecology of Invertebrates, Museum and Institute of Zoology Polish Academy of Science, Warsaw, Poland. [74]Key Laboratory of Urban Environment and Health, Institute of Urban Environment, Chinese Academy of Sciences, Xiamen, China. [75]Institute of Biology, Komi Science Centre, Ural Branch of Russian Academy of Sciences, Syktyvkar, Russia. [76]Institute of Ecology & Evolution, University of Bern, Bern, Switzerland. [77]Department of Ecology, School of Biology, Aristotle University of Thessaloniki, Thessaloniki, Greece. [78]Unaffiliated, Edmonton, Canada. [79]Canadian Forest Service, Natural Resources Canada, Sault Ste. Marie, Canada. [80]Department of Ecology, Swedish University of Agricultural Sciences, Uppsala, Sweden. [81]Greensway AB, Uppsala, Sweden. [82]Institute of Wildlife Management and Wildlife Biology, University of Sopron, Sopron, Hungary. [83]Key laboratory of Wetland Ecology and Environment, Northeast Institute of Geography and Agroecology, Chinese Academy of Sciences, Changchun 130102, China. [84]Key Laboratory of Vegetation Ecology, Ministry of Education, Northeast

Normal University, Changchun 130024, China. [85]Jilin Provincial Key Laboratory of Animal Resource Conservation and Utilization, Northeast Normal University, Changchun 130117, China. [86]Community Department, Helmholtz Center for Environmental Research, Halle, Germany. [87]Department of Biology, Paraiba State University, Campina Grande, Brazil. [88]Centre of Biodiversity and Sustainable Land Use, University of Göttingen, Göttingen, Germany.
✉e-mail: anton.potapov@biologie.uni-goettingen.de

