## [Peer Review File · Nature Communications]

Globally invariant metabolism but density-diversity mismatch in springtailsREVIEWER COMMENTS

Reviewer #1 (Remarks to the Author):

Dear Authors,

My compliments for your valuable work and your noteworthy results. According to me, your research will be of significance to soil ecology and provides additional value to comparable established literature on the macroecology of nematodes and earthworms, making your work is original. Your results support the well-written conclusions and are, as expected, closely related to similar patterns for soil nematodes. About your claims, please do not overestimate the springtails in comparison to free-living nematodes, which occupy three trophic levels and are much more relevant for biodiversity and biomass than Collembola. The random forest model is very appropriate and I agree with your interpretation, although I disagree with the vision that NDVI is a proxy for vegetation richness (living biomass yes, plant richness absolutely not). The methodology is quite sound although I regret that so important soil nutrients like nitrogen and phosphorus have been completely ignored as variables/predictors. Papers on soil food-web allometry clearly show that larger organisms like those belonging to the mesofauna (i.e., springtails) are highly sensitive to pH, N and P. Possibly something for a future approach, where ecological stoichiometry should play a major role (it is actually really THE driver). The methods are detailed enough to reproduce your computational work and the additional figures are very relevant.

Reviewer #2 (Remarks to the Author):

The manuscript deals with collembolan communities worldwide, their distribution, and global estimates for their biomass. This comes after a series of similar studies for other taxonomic groups as a global effort to unveil the biodiversity in soils. The results are an immense advancement of the field as it fills gaps in current knowledge at the global level. This work is original and will add to the background for a bigger goal of understanding how soil food webs work globally. Well-resolved food webs can increase our understanding of the processes that influence species diversity, ecosystem productivity, stability, as well as nutrient cycling.

I will start by saying Nature Communications is an appropriate journal for such an extensive and well-performed study like this. I will also mention that it was a pleasure reading this manuscript. The manuscript is well written and easy to follow and the aims of the study are clear. I have a few points for discussion:

1. The list of experts that vetted which data were included in the analyses could be more explicit as it is part of the methodology.
2. Given this is a large dataset, probably with graduate and/or undergraduate students collecting, analyzing and/or writing up the data, I ask: in detail, what were the authorship criteria? The rules need to be firmly established to be fair to everyone involved.
3. Nowhere in the main text the timing of the samples is mentioned, although the reporting summary does so. It may be worth adding that to your methods too, as this is part of the sampling effort for your analyses.
4. I know the point was not to compile a worldwide checklist of collembolan species, but it is natural to be interested in your final morpho-/species list. Site-level database of the #GlobalCollembola initiative is provided in FigShare, but not a species list. Is the list available somewhere else?
5. Please be clear where you used soil temperature vs. air temperature in your models.

Specifically:

Line 132: Springtails (Hexapoda: Collembola)

Line 189: While I understand that there are a lot of different terms across the document, being consistent would help the reader not to feel overwhelmed and feel that they need to be cross-referencing information between paragraphs, or in between text and figures. For example, in line 189 you mention that maximum densities were found in 'tundra', but in the abstract, you mention those happened in the 'Arctic'. However, the Arctic technically also comprises taiga ecosystems.

Line 475: Please state the maximum diameter of the areas to be considered independent sites as this reads vague.

Line 494: How would the 'no sample-level data' look like? Do you mean a checklist of species? Please expand.

Line 503: So, were all morphospecies at the genus level?

Line 516: Does the averaging mentioned here also account for time in the case of long-term studies?

Line 546: Do you anticipate any incongruences in your results between the analyses considering that the number of sites excluded per analysis differed?

Line 637: Please clarify what was considered 'grasslands' here as this reads confusing.

Extended Data Table 1: What is considered small vs. large for Isotomidae?

Reviewer #3 (Remarks to the Author):

Thanks for a really nice macroecological/biogeographical piece. I'm not that familiar with springtails, but it seems like a great bit of work, which has had a lot of effort put into it. I was actually quite impressed with the effort to gather and curate the data used in this analysis. This is never an easy task and looks to be a large collaborative effort. The figures are great and I love fig 1. I think that fig. 1a should become a stand alone image.

Major Comments

I do have a few methodological concerns, especially using a rarefied richness model, and calculating biomass from the average body mass equation. I think a paper that just used the 'raw' data, such as observed richness would be great, if the author's account for effort in number of samples and plug that directly into a richness model (poisson/negbin) then I would be less concerned about artefacts that can arise from rarefaction (such as inflated richness due to large number of rare species - often caused by undersampling bias; Coddington et al., 2009). I am slightly concerned with the richness prediction/model fits (often some of the richest sites are in temperate/arctic areas - fig 1 c), which seems to contradict the density vs richness argument. Would it be possible to see the raw richness data (as I understand it local species richness is calculated from Chao).

Maybe a map of effort is needed, number of sites per- region/cell? And then one on the number of samples (where known).

I think the role of seasonality/phenology should be explored a little more in the model fitting methods. The authors claim that they have dealt with this issue, but it seems to me the best thing to do would be to include all available data and have a seasonal effect in the models (rather than choosing samples at their seasonal density/diversity peak). Or include a covariate that represents seasonal climate variability.

I'd also think some type of sensitivity analysis needs to be done on the density/metabolism metrics, as these are derived from the data (not observed) choices in model coefficients (fits) could have a dramatic influence on the ratio between density/richness.

References

Coddington, J.A., Agnarsson, I., Miller, J.A., Kuntner, M. and Hormiga, G., 2009. Undersampling bias: the null hypothesis for singleton species in tropical arthropod surveys. *Journal of animal ecology*, pp.573-584.

Minor Comments

Line 108 - "megatonnes (Mt) of carbon" -Spell out this abbreviation.

Line 127: Maybe remove the “, predict,” from this sentence. I see prediction as part of the knowledge on where springtails are (a prediction).

Line 139: Remove multitrophic? Either is fine, but reads a bit better as biodiversity.

Line 167: “... here, we predicted “I think this needs to be reworded in species distribution literature. “Joint projection/prediction” has a very specific meaning (using a joint model). This paper uses community ecology stats - which is something different again.

Line 174: Super impressive dataset. Awesome that an initiative like this exists, and that ecologists and taxonomists are working together to understand the distribution of springtails.

Line 225: Yes, mean richness is higher in low latitudes, but looking at the raw data, there are clearly sites in Europe that are more species rich. Is there a reason for this? Like a difference in the sites over the less species rich ones (which there are plenty). I guess the point I’m getting to is there are random effects which can account for this difference? I.e. Agricultural vs forest? Or is this just a sampling artefact? More samples/more species?

Line 245: Yes looking at this result of Arctic richness and the supporting maps (ED Fig. 5). It actually looks as if high latitude areas are more/or as rich as the tropics. Which links to the point above. What is causing the low richness in North America, Europe ect?

For the richness analysis I wonder how much the rarefaction method is inflating/deflating richness in predictions. Looking at the maps of raw richness there doesn’t seem to be the same major trend (mind you the colour scale is a bit hard to tell). I’m wondering instead of fitting a rarefied richness if the authors could fit a Poisson/negative binomial model with an offset for effort (number of sites per sample). Something like this in R should do the trick:

```
glm(richness ~ covariates + offset(log(No.Samples)),family=poisson)
```

297: I wonder if there is a phenological signal in this data. Sounds like tundra are being sampled when springtail populations are large (during summer). When are the other sites sampled? Is there an effect of seasonality in the data? Might be worth checking.

Line 334-336: Starting with “This,...”. This sentence doesn’t really make sense to me and needs to be revisited.

Line 331: Are there operational taxonomic units to represent undescribed or unidentified species? Taxonomically resolving global taxon data bases is not easy, but it might help with consistency across species. The linnean and wallacean shortfalls are always an issue when it comes to biogeography/macroecology.

Line 336: It might be nice to explore how NDVI is projected to change with climate change? Maybe just from a qualitative sense (literature) is NDVI expected to increase or decrease in areas important for springtails? Do we expect to see both arctic tundra and areas of high NDVI decrease?

Methods

Line 470: Impressive dataset and even more impressive that the authors went through the primary literature to source new datasets. Will this data be made publicly available? Or at least a release to recreate these results.

Line 481-484: I realise the authors have said that they have dealt with the seasonal effect, but I wonder if this is true? Surely the best thing to do would be to include this as a factor in the model to see how richness changed depending on the season. It might even control for some of the latitudinal

patterns of density/richness if there is a strong seasonal effect.

Line 489-496: I am not a huge fan of rarefaction approaches, as I feel there are so many ways in which they can be biased (effort/rarity ect). But I totally understand this is a widely used approach. So I wonder if the same models could be built for just the observed richness/biomass data and an offset included as described above. You're basically doing something similar with the completeness estimates, but rather than a two step approach (rarefied richness, completeness), you could just do a richness model on observed data.

Line 503-508: Do you not have a direct measure of biomass? But rather are calculating it from body length/traits? I imagine this would be quite sensitive to the coefficients used to fit this relationship in the average body mass equation. This is especially true with exponents which can have drastic change on outcomes with small changes in coefficient values. A sensitivity analysis to show that this is not drastically altering the richness/density relationship might be warranted. Or perhaps a map/prediction of average body length, are springtails in general larger in arctic regions? Sometimes bodysize can follow different patterns in extreme/different environments. This can happen in island samples, and places like the deep-sea/polar regions. These larger body sizes are due to resource limitation, this is called the island rule: Foster JB (1964) Evolution of mammals on islands. Nature 202: 234–235.

Line 598: Probably need a little more on this section, how well did the RF models do? I see you did model selection based on R^2 , but it is often better to use some type of information criteria, like BIC, which can penalise models with similar log likelihoods but are more heavily parameterised. You also need to report these in the supporting materia/extended data. As it's a bit hard to take model selection on face value. Did you do cross validation, predictive tests (AUC)?

I like the addition of the bootstrapping, and reporting of uncertainty in predictions for the extended data.

Lines 630-641: I'm not sure why you are fitting a SEM here? There are no latent variables in the model and it feels like fitting a regression/glm would give you the same thing. The steps to doing the path analysis seem sound, checking for collinearity and splitting the data in IPBES regions. But surely a glm would do the same thing? And it would have the added benefit of model selection and diagnostics (residuals) - which are missing.

Figures:

Just a minor thing, but it's easier to reference the figures in the order they enter the text. So d->c and g ->d. Ect.

Fig. 1: The N samples legend/points are a bit hard to differentiate at this size. Maybe the size and colour could help.

Fig. 2. Love this map. I would almost make Fig. 2a a separate figure. There is a log of information to digest here.

REVIEWER COMMENTS AND OUR RESPONSES

In our responses we refer to the lines in the clean version of the revised manuscript.

Reviewer #1 (Remarks to the Author):

Dear Authors,

My compliments for your valuable work and your noteworthy results. According to me, your research will be of significance to soil ecology and provides additional value to comparable established literature on the macroecology of nematodes and earthworms, making your work is original. Your results support the well-written conclusions and are, as expected, closely related to similar patterns for soil nematodes. About your claims, please do not overestimate the springtails in comparison to free-living nematodes, which occupy three trophic levels and are much more relevant for biodiversity and biomass than Collembola. The random forest model is very appropriate and I agree with your interpretation, although I disagree with the vision that NDVI is a proxy for vegetation richness (living biomass yes, plant richness absolutely not). The methodology is quite sound although I regret that so important soil nutrients like nitrogen and phosphorus have been completely ignored as variables/predictors. Papers on soil food-web allometry clearly show that larger organisms like those belonging to the mesofauna (i.e., springtails) are highly sensitive to pH, N and P. Possibly something for a future approach, where ecological stoichiometry should play a major role (it is actually really THE driver). The methods are detailed enough to reproduce your computational work and the additional figures are very relevant.

Response: Thank you for this positive evaluation of our work. We agree that trophic diversity and global activity are higher for nematodes than for springtails.

Nevertheless, our estimates show that springtails contribute similarly to the global biomass. They also play an engineering role in soil and serve as important agents connecting below- and aboveground food webs, because they serve as prey for major generalist predators. Moreover, springtails also span across three trophic levels (<https://doi.org/10.1016/j.soilbio.2005.02.006>). We agree that a comparison of significance of these two groups of organisms is not feasible and useful, but that both are critical players in soil food webs and ecosystem functioning. We now have avoided any statements that overestimate springtail 'importance' in comparison to e.g. free-living nematodes. Instead, we provide our estimates in the context of other relevant animal groups and total soil respiration.

According to your suggestion, we reworded the interpretation of NDVI from 'richness' to 'biomass'.

While the global distribution of soil nitrogen has been described (<https://soilgrids.org>), it is a modelled product, which strongly correlates with soil carbon and thus cannot be used as an independent predictor. Unfortunately, we cannot include soil phosphorus in our global models, because these data are not available at the global scale. We agree that including stoichiometry in future global models on soil biodiversity, especially across size classes, would be an important step towards understanding the soil biosphere. However, this is beyond the scope of our paper. We now provide these explanations in the main text (LL 283-286) and in

the methods (LL 639-643) and state in the outlook that future models should include independent assessments of soil carbon and nitrogen.

Reviewer #2 (Remarks to the Author):

The manuscript deals with collembolan communities worldwide, their distribution, and global estimates for their biomass. This comes after a series of similar studies for other taxonomic groups as a global effort to unveil the biodiversity in soils. The results are an immense advancement of the field as it fills gaps in current knowledge at the global level. This work is original and will add to the background for a bigger goal of understanding how soil food webs work globally. Well-resolved food webs can increase our understanding of the processes that influence species diversity, ecosystem productivity, stability, as well as nutrient cycling.

I will start by saying Nature Communications is an appropriate journal for such an extensive and well-performed study like this. I will also mention that it was a pleasure reading this manuscript. The manuscript is well written and easy to follow and the aims of the study are clear. I have a few points for discussion:

1. The list of experts that vetted which data were included in the analyses could be more explicit as it is part of the methodology.

Response: Thank you for the valuation of our global synthesis. The list of experts is now included in the supplementary “Data cleaning protocol”. Here it is: Anatoly Babenko – high latitude regions in both north and south hemispheres, Bruno Bellini – Central and South America, Jean-François Ponge – Central and Western Europe, Louis Deharveng – Africa and Asia, Lubomir Kovac – Southern Europe, Mikhail Potapov and Natalia Kuznetsova – Eastern and Northern Europe. This information and evaluation grades will also be included in a detailed version of the database which will be published soon after the present synthesis study.

2. Given this is a large dataset, probably with graduate and/or undergraduate students collecting, analyzing and/or writing up the data, I ask: in detail, what were the authorship criteria? The rules need to be firmly established to be fair to everyone involved.

Response: Indeed, this is an important topic. We developed an authorship strategy which was communicated to all data providers during the data-collection phase. All direct data providers who collected and standardised the data were invited as co-authors. For unpublished data, people who were directly involved in sorting and identification of springtails were invited as co-authors. Principal investigators were NOT included as co-authors, unless the research project was focused on springtails, and consequently these PIs contributed to conceptualisation/writing of the manuscript. Information on our co-authorship guidelines was added to the revised version of the manuscript (LL 487-489).

3. Nowhere in the main text the timing of the samples is mentioned, although the reporting summary does so. It may be worth adding that to your methods too, as this is part of the sampling effort for your analyses.

Response: Agreed. We now provide this information in the dataset description in the methods (LL 604-607): “The dataset covered all major biomes (Extended Data Fig. 3), years 1970-2019, and all months: 8% of the samples were taken between December and February, 14% between March and May, 55% between June and August, and 23% between September and November.”

4. I know the point was not to compile a worldwide checklist of collembolan species, but it is natural to be interested in your final morpho-/species list. Site-level database of the #GlobalCollembola initiative is provided in FigShare, but not a species list. Is the list available somewhere else?

Response: In the framework of the #GlobalCollembola activities, we mobilised the global checklist of Collembola species available from www.collembola.org. We structured taxonomic tables and shared them with the Catalogue of Life (<https://www.catalogueoflife.org>), and now the checklist version from 2020 is included in GBIF (<https://gbif.org>). However, we only validated the global community database at the genus level (with all names and synonyms checked) because accurate body size information at the species-level was not available. Taxonomy and trait databases are work in progress. To substantiate the present paper, we have deposited site- and event-level data in Figshare. A detailed sample-based version of the database with validated species names will be published soon after the present synthesis study.

5. Please be clear where you used soil temperature vs. air temperature in your models.

Response:

We now explicitly specify ‘air temperature’ and ‘soil temperature’ across the text. Springtail metabolism was calculated using soil temperatures, because this is what influences this parameter directly. Extrapolations, SEM, and linear modelling were done with air temperatures, as this is the conventional parameter used across geospatial analyses.

Specifically:

Line 132: Springtails (Hexapoda: Collembola)

Response: Corrected.

Line 189: While I understand that there are a lot of different terms across the document, being consistent would help the reader not to feel overwhelmed and feel that they need to be cross-referencing information between paragraphs, or in between text and figures. For example, in line 189 you mention that maximum

densities were found in 'tundra', but in the abstract, you mention those happened in the 'Arctic'. However, the Arctic technically also comprises taiga ecosystems.

Response: We now replaced 'the Arctic' in the abstract with 'tundra'. However, 'the Arctic' is mentioned two times in the text as a geographical region. Moreover, addressing this comment, we carefully checked the whole manuscript for consistent terminology during revisions.

Line 475: Please state the maximum diameter of the areas to be considered independent sites as this reads vague.

Response: Unfortunately, we don't have information on the sampling 'site' size for each study. The spatial scope of the sampling and community boundaries is a common and poorly-explored issue in soil ecology. Traditionally, Collembola assessments are done within plots of 10-50 m in diameter. Since we collected existing data across many years and studies, we can only provide a rough estimation. We modified the text as follows (LL 505-508): "Here, we defined a site as a locality that hosts a defined springtail community, is covered by a certain vegetation type, with a certain management, and is usually represented by a sampling area of up to a hundred metres in diameter, making species co-occurrence and interactions plausible."

Line 494: How would the 'no sample-level data' look like? Do you mean a checklist of species? Please expand.

Response: Many studies report only site-level averages on density and species richness, no raw data on species identity and numbers. We now clarified this in the text (LL 521-523).

Line 503: So, were all morphospecies at the genus level?

Response: Yes, in almost all cases. Where genus-level identification was not available, we excluded sites from the analysis. This information was added to the revised manuscript (L 537).

Line 516: Does the averaging mentioned here also account for time in the case of long-term studies?

Response: No, we used mean annual soil temperature. The seasonal effects are now explored in the new "Linear mixed-effect modelling" analysis (Extended Data Fig. 10).

Line 546: Do you anticipate any incongruences in your results between the analyses considering that the number of sites excluded per analysis differed?

Response: Indeed, this might have introduced some incongruences between the analyses. However, most of the sites reported all parameters. Moreover, all direct comparisons (correlations) were done using the same sites and thus these incongruences do not have any impacts on our main conclusions.

Line 637: Please clarify what was considered 'grasslands' here as this reads confusing.

Response: We now clarified the basis of the classification and how the habitat was coded in the model (LL 703-706): '...vegetation cover reported by the data providers following the habitat classification of European Environment Agency (woodland, scrub, agriculture, or grasslands; the latter were coded as the combination of woodland, scrub, and agriculture absent).'

Extended Data Table 1: What is considered small vs. large for Isotomidae?

Response: The threshold between small and large Isotomidae has been set at 1.5 mm according to the size ranges of the common genera and in account for the chosen regression coefficients (*Folsomia* and *Isotomiella* = small; *Desoria* and *Isotoma* = large). We now added the size threshold to the table.

Reviewer #3 (Remarks to the Author):

Thanks for a really nice macroecological/biogeographical piece. I'm not that familiar with springtails, but it seems like a great bit of work, which has had a lot of effort put into it. I was actually quite impressed with the effort to gather and curate the data used in this analysis. This is never an easy task and looks to be a large collaborative effort. The figures are great and I love fig 1. I think that fig. 1a should become a stand alone image.

Response: Thank you for the valuation of our collaborative effort and figures. We believe that Fig. 1a is rather methodological (it shows the distribution of sampling sites) and would prefer to keep it in conjunction with the panels b-g, which show data analyses. This way, we are able to present our story more concisely and focus the reader's attention on the research results.

Major Comments

I do have a few methodological concerns, especially using a rarefied richness model, and calculating biomass from the average body mass equation. I think a paper that just used the 'raw' data, such as observed richness would be great, if the author's account for effort in number of samples and plug that directly into a richness model (poisson/negbin) then I would be less concerned about artefacts that can arise from rarefaction (such as inflated richness due to large number of rare species - often caused by undersampling bias; Coddington et al., 2009). I am slightly concerned with

the richness prediction/model fits (often some of the richest sites are in temperate/arctic areas - fig 1 c), which seems to contradict the density vs richness argument. Would it be possible to see the raw richness data (as I understand it local species richness is calculated from Chao).

Response: Since our database is compiled based on many studies that differ in sampling effort and soil sample size, we believe that extrapolation of species richness is a necessary step for standardisation. To the best of our knowledge and data, the issue of singletons, even in tropical springtail communities, is not as severe as in aboveground tropical arthropod communities. To test if raw richness data would lead to different conclusions, we implemented additional linear mixed-effect modelling with “Total collection area” and “Sampled microhabitats” (only soil, only litter, and both) as additional predictors (LL 729-743). The results are presented in Extended Data Fig 10 and discussed in the main text (LL 309-315 and 338-340). The drivers of raw species richness are similar to that of extrapolated species richness, and the correlations with density and metabolism were also weak (all R^2 values were between 0.02 and 0.07; LL 249-250).

Maybe a map of effort is needed, number of sites per- region/cell? And then one on the number of samples (where known).

Response: Agreed. We provide information on the sampling efforts in terms of the sampling-site distribution in Fig. 1a, methods, and Extended Data Figs 1, 2 and 3. Information on the number of samples is also displayed in Fig. 1a. We produced additional density-per-cell maps for the number of sites and samples which are now presented in Extended Data Fig. 2.

I think the role of seasonality/phenology should be explored a little more in the model fitting methods. The authors claim that they have dealt with this issue, but it seems to me the best thing to do would be to include all available data and have a seasonal effect in the models (rather than choosing samples at their seasonal density/diversity peak). Or include a covariate that represents seasonal climate variability.

Response: We agree that seasonality/phenology is an important aspect affecting community parameters in some regions. It is a poorly-explored and difficult issue in the global modelling of soil biodiversity. We had to focus on the vegetation biomass peak periods globally, because this is what our data represent. Covariates which are related to climate variability (temperature seasonality, temperature annual range, precipitation seasonality, precipitation of the driest quarter) were included in both random forest and structural equation modelling (LL 633-639). However, they were not selected for the final structural equation model due to collinearity with other predictors.

To further explore the potential seasonality bias, we now built additional linear mixed-effects models using only those sampling events where the sampling year and month were known (69% of all sites). We linked these data to specific year- and month-based data on air temperature and precipitation (CHELSA) and used ‘Site’ as random effect to account for interdependence of the events coming from the same site. Monthly temperature was analysed as offset from the annual mean for the given

site to avoid collinearity with the latter (LL 729-743). Results of this modelling showed that seasonal effects of climate are generally smaller than the effects of annual climatic means (see Extended Data Fig 10). However, while analysing sampling events, we discovered very strong effects of mean annual and monthly temperatures on community metabolism. This implies high variation of this community metric with temperature. We believe this is an interesting avenue for future research and describe this in the main text (LL 309-315).

I'd also think some type of sensitivity analysis needs to be done on the density/metabolism metrics, as these are derived from the data (not observed) choices in model coefficients (fits) could have a dramatic influence on the ratio between density/richness.

Response: Indeed, indirect estimations of the biomass and metabolism are sensitive to the used coefficients. To estimate the potential bias, we now used reported standard errors of coefficients in the length-mass and mass-metabolic rate allometric equations to calculate 'maximum' (mean coefficient + standard error) and 'minimum' (mean coefficient - standard error) biomass and community metabolism for each site (LL 548-552). We then run again the global extrapolations to include these uncertainties in our global estimates. The results are given in the text (LL 359-362): our global biomass estimates averaged 27.5 Mt C with 16.2 as the minimum and 28.8 Mt C as the maximum; our global community metabolism estimates averaged 15.2 Mt C month⁻¹ with 14.6 as the minimum and 18.6 Mt C month⁻¹ as the maximum.

References

Coddington, J.A., Agnarsson, I., Miller, J.A., Kuntner, M. and Hormiga, G., 2009. Undersampling bias: the null hypothesis for singleton species in tropical arthropod surveys. *Journal of animal ecology*, pp.573-584.

Minor Comments

Line 108 - "megatonnes (Mt) of carbon" -Spell out this abbreviation.

Line 127: Maybe remove the ", predict," from this sentence. I see prediction as part of the knowledge on where springtails are (a prediction).

Line 139: Remove multitrophic? Either is fine, but reads a bit better as biodiversity.

Response: All done.

Line 167: "... here, we predicted "I think this needs to be reworded in species distribution literature. "Joint projection/prediction" has a very specific meaning (using a joint model). This paper uses community ecology stats - which is something different again.

Response: We removed the 'joint'.

Line 174: Super impressive dataset. Awesome that an initiative like this exists, and that ecologists and taxonomists are working together to understand the distribution of springtails.

Response: Thank you.

Line 225: Yes, mean richness is higher in low latitudes, but looking at the raw data, there are clearly sites in Europe that are more species rich. Is there a reason for this? Like a difference in the sites over the less species rich ones (which there are plenty). I guess the point I'm getting to is there are random effects which can account for this difference? I.e. Agricultural vs forest? Or is this just a sampling artefact? More samples/more species?

Response: We believe that this is explained mainly by the sampling density. European temperate ecosystems are widely studied, because historically the taxonomic expertise concentrated in Europe. So, existing surveys observed both species-rich and species-poor communities. The pattern is unlikely to be explained by the habitat. For example, the global gradient for 'woodlands' (shown below) resembles the total gradient. We discuss this issue in the LL 350-358.

Line 245: Yes looking at this result of Arctic richness and the supporting maps (ED Fig. 5). It actually looks as if high latitude areas are more/or as rich as the tropics. Which links to the point above. What is causing the low richness in North America, Europe ect?

Response: Indeed, species-rich communities are observed and predicted to exist both at low and high latitudes. In turn, temperate Europe and North America have large areas with moderate predicted local species richness. These regions are highly transformed and covered by large agricultural / crop areas; despite the overall moderate negative effect of agriculture on springtail species richness, such transformations may have depleted regional-level diversity. We now added this consideration to the discussion of our results (LL 336-338): "...negative effects of agriculture and other human activities are supported by the moderate predicted local species richness in many areas of highly transformed landscapes in Europe and North America (Fig. 2)."

For the richness analysis I wonder how much the rarefaction method is inflating/deflating richness in predictions. Looking at the maps of raw richness there doesn't seem to be the same major trend (mind you the colour scale is a bit hard to tell). I'm wondering instead of fitting a rarefied richness if the authors could fit a Poisson/negative binomial model with an offset for effort (number of sites per sample). Something like this in R should do the trick:

```
glm(richness ~ covariates + offset(log(No.Samples)),family=poisson)
```

Response: We ran additional generalised linear mixed-effect models to predict raw species richness with "Total collection area" and "Sampled microhabitats" (only soil, only litter, and both) as additional predictors (LL 729-743). The results are now presented in Extended Data Fig 10 and discussed in the text (LL 355-358). The drivers of raw species richness are similar to those of extrapolated species richness.

297: I wonder if there is a phenological signal in this data. Sounds like tundra are being sampled when springtail populations are large (during summer). When are the other sites sampled? Is there an effect of seasonality in the data? Might be worth checking.

Response: We had to focus on the vegetation biomass peak periods globally, because this is what our data represent. We now provide information on the temporal coverage of our data in the methods (LL 604-607): "The dataset covered all major biomes (Extended Data Fig. 3), years 1970-2019, and all months: 8% of samples were taken between December and February, 14% between March and May, 55% between June and August, and 23% between September and November." To further explore the potential seasonality bias, we now built additional linear mixed-effects models using only those sampling events, where the sampling year and month were known (see our answers above).

Line 334-336: Starting with "This,...". This sentence doesn't really make sense to me and needs to be revisited.

Response: We now start the sentence with "Hence, ...".

Line 331: Are there operational taxonomic units to represent undescribed or unidentified species? Taxonomically resolving global taxon data bases is not easy, but it might help with consistency across species. The linnean and wallacean shortfalls are always an issue when it comes to biogeography/macroecology.

Response: We included several datasets from tropical countries that are partly based on morphospecies. At present, this is a common way to implement springtail community research in the tropics. Identification to the genus level is possible in most cases. Although delineation below the genus level is not trivial, a common set of family/genus specific identification characters is used by specialists, allowing to make these delineations more comparable. All morphospecies datasets were produced by experienced specialists and validated by our expert team (Extended Data Fig. 1). Some of the morphospecies are also available from a morphospecies repository, e.g. <http://ecotaxonomy.org/taxa/407469>. Such data would not allow us to compare community composition, but are suitable to adequately assess species richness. We specify our approach in LL 518-519 and 534-537.

Line 336: It might be nice to explore how NDVI is projected to change with climate change? Maybe just from a qualitative sense (literature) is NDVI expected to increase or decrease in areas important for springtails? Do we expect to see both arctic tundra and areas of high NDVI decrease?

Response: Thank you for this interesting suggestion. However, in the current project, we were interested in representing the current situation as best as possible, and did not focus on future changes. NDVI is a remotely-sensed product, and, to our knowledge, no high-quality future predictions exist. Future projections of NDVI and/or springtail communities would certainly be very interesting follow-up projects. We now mention this briefly in the perspectives section of our paper (L 373).

Methods

Line 470: Impressive dataset and even more impressive that the authors went through the primary literature to source new datasets. Will this data be made publicly available? Or at least a release to recreate these results.

Response: Site- and now also event-level datasets are openly available from Figshare under CC-BY 4.0 license: <https://doi.org/10.6084/m9.figshare.16850419>. A detailed sample-based version of the database with validated species names will be published and openly available soon after the present synthesis study.

Line 481-484: I realise the authors have said that they have dealt with the seasonal effect, but I wonder if this is true? Surely the best thing to do would be to include this as a factor in the model to see how richness changed depending on the season. It might even control for some of the latitudinal patterns of density/richness if there is a strong seasonal effect.

Response: This is indeed an important point. We now explicitly explore seasonal effects in additional models (see our answers above).

Line 489-496: I am not a huge fan of rarefaction approaches, as I feel there are so many ways in which they can be biased (effort/rarity ect). But I totally understand this is a widely used approach. So I wonder if the same models could be built for just the observed richness/biomass data and an offset included as described above. You're basically doing something similar with the completeness estimates, but rather than a two step approach (rarefied richness, completeness), you could just do a richness model on observed data.

Response: To address this comment, we now explicitly model raw richness data (see our answers above).

Line 503-508: Do you not have a direct measure of biomass? But rather are calculating it from body length/traits? I imagine this would be quite sensitive to the coefficients used to fit this relationship in the average body mass equation. This is especially true with exponents which can have drastic change on outcomes with small changes in coefficient values. A sensitivity analysis to show that this is not drastically altering the richness/density relationship might be warranted. Or perhaps a map/prediction of average body length, are springtails in general larger in arctic regions? Sometimes bodysize can follow different patterns in extreme/different environments. This can happen in island samples, and places like the deep-sea/polar regions. These larger body sizes are due to resource limitation, this is called the island rule: Foster JB (1964) Evolution of mammals on islands. Nature 202: 234–235.

Response: Unfortunately, community or population biomass data of springtails are scarce. We now added extrapolations that include coefficient uncertainties, and these results are presented in the text (LL 359-362; see our answers above). We started the collection of a global trait dataset on springtails, which represents another massive data mobilisation and synthesis project. We plan to implement a detailed body size distribution analysis in the framework of this project. We prefer not to show a map with more rough estimations at this stage.

Line 598: Probably need a little more on this section, how well did the RF models do? I see you did model selection based on R^2 , but it is often better to use some type of information criteria, like BIC, which can penalise models with similar log likelihoods but are more heavily parameterised. You also need to report these in the supporting materia/extended data. As it's a bit hard to take model selection on face value. Did you do cross validation, predictive tests (AUC)?

Response: Agreed. Each of our models was tested using cross-validation (with folds assigned randomly), and the final prediction is presented as an ensemble of 10 models. The coefficient of determination (R^2) was used to select the top 10 best-performing models for the final ensemble (LL 652-654). We believe that for this type

of model (Random Forest regression), these model statistics are more appropriate than BIC and/or AIC. Additionally, we now included results from a spatial leave-one-out procedure (see Roberts et al. 2016

<https://onlinelibrary.wiley.com/doi/10.1111/ecog.02881> and Ploton et al. 2021 <https://www.nature.com/articles/s41467-020-18321-y>) that shows that our models have very small spatial autocorrelation and a reasonable predictive power, even when leaving out data from nearby locations. We describe these new statistics in the methods (LL 658-668) and also included a new supplementary file 'Extrapolation spatial validation' with a Figure.

I like the addition of the bootstrapping, and reporting of uncertainty in predictions for the extended data.

Response: Thank you.

Lines 630-641: I'm not sure why you are fitting a SEM here? There are no latent variables in the model and it feels like fitting a regression/glm would give you the same thing. The steps to doing the path analysis seem sound, checking for collinearity and splitting the data in IPBES regions. But surely a glm would do the same thing? And it would have the added benefit of model selection and diagnostics (residuals) - which are missing.

Response: We are using a conventional SEM approach based on a covariance matrix, which allowed us to estimate direct and indirect effects of several interdependent factors. We now added linear mixed effects modelling as an alternative 'sensitivity' analysis based on sampling events (as described below).

Figures:

Just a minor thing, but it's easier to reference the figures in the order they enter the text. So d->c and g ->d. Ect.

Response: Order of the letters was changed accordingly.

Fig. 1: The N samples legend/points are a bit hard to differentiate at this size. Maybe the size and colour could help.

Response: We agree that it is hard to differentiate the exact number of samples for each point from the map. However, most of the sites have similar sampling effort (5-10 samples per site; see the histogram below), and our map illustrates this. We believe that stretching the size scale (see the map below) and adding colours will not improve clarity. To provide information in a more clear form, we added a map representing the density of sampling efforts in Extended Data Fig. 2.

Fig. 2. Love this map. I would almost make Fig. 2a a separate figure. There is a log of information to digest here.

Response: Thank you. We agree that the map (Fig. 2a) qualifies for a separate figure. However, the two maps (panels a and b) present information in a similar way. This is why we prefer this more intuitive presentation.

REVIEWER COMMENTS

Reviewer #2 (Remarks to the Author):

I was reviewer #2 and I am happy with how you addressed the concerns we all had. Specifically, thanks for clarifying the authorship issue that we from the Southern Hemisphere all know exists and favors bias. Best of luck, and congratulations for the massive effort.

Reviewer #3 (Remarks to the Author):

First of all, I would like to sincerely apologise for the slowness of my review. I think the authors have done a great job at responding to my original comments, the paper is greatly improved and it is a lovely piece of work that should be seriously considered for publication. I only had a few re-responses to previous comments, none of them should take too much extra effort to address.

Re-responses

Previous comment: I do have a few methodological concerns, especially using a rarefied richness model, and calculating biomass from the average body mass equation. I think a paper that just used the 'raw' data, such as observed richness would be great, if the author's account for effort in number of samples and plug that directly into a richness model (poisson/negbin) then I would be less concerned about artefacts that can arise from rarefaction (such as inflated richness due to large number of rare species - often caused by undersampling bias; Coddington et al., 2009). I am slightly concerned with 6 the richness prediction/model fits (often some of the richest sites are in temperate/arctic areas - fig 1 c), which seems to contradict the density vs richness argument. Would it be possible to see the raw richness data (as I understand it local species richness is calculated from Chao).

Response: Since our database is compiled based on many studies that differ in sampling effort and soil sample size, we believe that extrapolation of species richness is a necessary step for standardisation. To the best of our knowledge and data, the issue of singletons, even in tropical springtail communities, is not as severe as in aboveground tropical arthropod communities. To test if raw richness data would lead to different conclusions, we implemented additional linear mixed-effect modelling with "Total collection area" and "Sampled microhabitats" (only soil, only litter, and both) as additional predictors (LL 729-743). The results are presented in Extended Data Fig 10 and discussed in the main text (LL 309-315 and 338-340). The drivers of raw species richness are similar to that of extrapolated species richness, and the correlations with density and metabolism were also weak (all R2 values were between 0.02 and 0.07; LL 249-250).

Re-response: I really appreciate the additional analyses and am happy that extrapolation is not causing any undue biases in the results and inference. I really like the side comparison between raw and extrapolated richness in ED Fig. 10. Thanks.

Previous comment: Maybe a map of effort is needed, number of sites per- region/cell? And then one on the number of samples (where known).

Response: Agreed. We provide information on the sampling efforts in terms of the sampling-site distribution in Fig. 1a, methods, and Extended Data Figs 1, 2 and 3. Information on the number of samples is also displayed in Fig. 1a. We produced additional density-per-cell maps for the number of sites and samples which are now presented in Extended Data Fig. 2.

Re-response: Thanks for doing this. I really like this plot. I'd ask for one minor change and that is to report the results on the natural log scale. This will help reduce the influence of the few cells with very high sites/samples and hopefully identify any pattern (or none).

Previous comment: I think the role of seasonality/phenology should be explored a little more in the model fitting methods. The authors claim that they have dealt with this issue, but it seems to me the best thing to do would be to include all available data and have a seasonal effect in the models (rather than choosing samples at their seasonal density/diversity peak). Or include a covariate that represents seasonal climate variability.

Response: We agree that seasonality/phenology is an important aspect affecting community parameters in some regions. It is a poorly-explored and difficult issue in the global modelling of soil biodiversity. We had to focus on the vegetation biomass peak periods globally, because this is what our data represent. Covariates which are related to climate variability (temperature seasonality, temperature annual range, precipitation seasonality, precipitation of the driest quarter) were included in both random forest and structural equation modelling (LL 633-639). However, they were not selected for the final structural equation model due to collinearity with other predictors. To further explore the potential seasonality bias, we now built additional linear mixed-effects models using only those sampling events where the sampling year and month were known (69% of all sites). We linked these data to specific year- and month-based data on air temperature and precipitation (CHELSA) and used 'Site' as random effect to account for interdependence of the events coming from the same site. Monthly temperature was analysed as offset from the annual mean for the given 7 site to avoid collinearity with the latter (LL 729-743). Results of this modelling showed that seasonal effects of climate are generally smaller than the effects of annual climatic means (see Extended Data Fig 10). However, while analysing sampling events, we discovered very strong effects of mean annual and monthly temperatures on community metabolism. This implies high variation of this community metric with temperature. We believe this is an interesting avenue for future research and describe this in the main text (LL 309-315).

Re-response: Great! I really appreciate this additional analysis, very interesting about the potential effect of climate variability on the community metabolism. It would be fascinating to explore that relationship under future climate scenarios. There seems to be a mild (confidence intervals not overlapping zero) positive effect of monthly temperature on richness/density, which might make sense if there is higher richness/density in places that have lower mean annual temperature [MAT] regions (and typically have greater seasonal variability - temperate vs tropical regions). Which can be seen in ED Fig 8 and 10. It also looks like the relationship between MAT and raw richness/extrapolated richness is negative when using a linear term (ED Fig. 10). This would suggest higher richness in colder environments and potentially contradict the latitudinal gradients patterns. I personally think this is just a case that a unimodal (quadratic form) would likely do a better job at explaining this relationship (as is shown in Fig. 2) than a linear form (ED Fig. 10); it might be worth revisiting this model with polynomials for each covariate.

Previous comment: I'd also think some type of sensitivity analysis needs to be done on the density/metabolism metrics, as these are derived from the data (not observed) choices in model coefficients (fits) could have a dramatic influence on the ratio between density/richness.

Response: Indeed, indirect estimations of the biomass and metabolism are sensitive to the used coefficients. To estimate the potential bias, we now used reported standard errors of coefficients in the length-mass and mass-metabolic rate allometric equations to calculate 'maximum' (mean coefficient + standard error) and 'minimum' (mean coefficient - standard error) biomass and community metabolism for each site (LL 548-552). We then run again the global extrapolations to include these uncertainties in our global estimates. The results are given in the text (LL 359-362): our global biomass estimates averaged 27.5 Mt C with 16.2 as the minimum and 28.8 Mt C as the maximum; our global community metabolism estimates averaged 15.2 Mt C month⁻¹ with 14.6 as the minimum and 18.6 Mt C month⁻¹ as the maximum

Re-response: This is a great addition, I think the authors for taking the time to redo these analyses and alleviate my previous concerns.

Previous comment: Line 225, Yes, mean richness is higher in low latitudes, but looking at the raw

data, there are clearly sites in Europe that are more species rich. Is there a reason for this? Like a difference in the sites over the less species rich ones (which there are plenty). I guess the point I'm getting to is there are random effects which can account for this difference? I.e. Agricultural vs forest? Or is this just a sampling artefact? More samples/more species?

Response: We believe that this is explained mainly by the sampling density. European temperate ecosystems are widely studied, because historically the taxonomic expertise concentrated in Europe. So, existing surveys observed both species-rich and species-poor communities. The pattern is unlikely to be explained by the habitat. For example, the global gradient for 'woodlands' (shown below) resembles the total gradient. We discuss this issue in the LL 350-358.

Re-response: Thanks for this explanation, and the discussion in the text.

Previous comment: For the richness analysis I wonder how much the rarefaction method is inflating/deflating richness in predictions. Looking at the maps of raw richness there doesn't seem to be the same major trend (mind you the colour scale is a bit hard to tell). I'm wondering instead of fitting a rarefied richness if the authors could fit a Poisson/negative binomial model with an offset for effort (number of sites per sample). Something like this in R should do the trick: `glm(richness ~ covariates + offset(log(No.Samples)),family=poisson)`

Response: We ran additional generalised linear mixed-effect models to predict raw species richness with "Total collection area" and "Sampled microhabitats" (only soil, only litter, and both) as additional predictors (LL 729-743). The results are now presented in Extended Data Fig 10 and discussed in the text (LL 355-358). The drivers of raw species richness are similar to those of extrapolated species richness

Re-response: Great, I appreciate the addition of raw species richness in the analyses. I was pointing towards the idea of tampering richness by effort (eg. using an offset). Rarefaction is trying to do this to some extent. You could think about in terms of say fisheries data, where they often use catch-per-unit-effort. This will give you the "corrected" count/biomass/weight of a species/community based on a relationship between effort (how many trawls/samples) and catch (what you get in your samples). There are a bunch of other issues (like does this relationship scale linearly, is effort biased towards higher numbers of the species), but it typically helps reduce bias in raw numbers and gives a better perspective of species numbers in relation to the effort used to collect/fish the target species. I'd like to make it clear that I don't expect the authors to do any additional analyses here, but I was just pointing out how that might be useful for future analyses.

REVIEWERS' COMMENTS

Reviewer #2 (Remarks to the Author):

I was reviewer #2 and I am happy with how you addressed the concerns we all had. Specifically, thanks for clarifying the authorship issue that we from the Southern Hemisphere all know exists and favors bias. Best of luck, and congratulations for the massive effort.

Response: Thank you. We now included “Inclusion & Ethics” section in the methods where we provide further details on the communication and inclusiveness of the initiative.

Reviewer #3 (Remarks to the Author):

First of all, I would like to sincerely apologise for the slowness of my review. I think the authors have done a great job at responding to my original comments, the paper is greatly improved and it is a lovely piece of work that should be seriously considered for publication. I only had a few re-responses to previous comments, none of them should take too much extra effort to address.

Response: Thank you for your detailed comments. We provide our final remarks/edits below, point-by-point.

Re-responses

Previous comment: I do have a few methodological concerns, especially using a rarefied richness model, and calculating biomass from the average body mass equation. I think a paper that just used the ‘raw’ data, such as observed richness would be great, if the author's account for effort in number of samples and plug that directly into a richness model (poisson/negbin) then I would be less concerned about artefacts that can arise from rarefaction (such as inflated richness due to large number of rare species - often caused by undersampling bias; Coddington et al., 2009). I am slightly concerned with 6 the richness prediction/model fits (often some of the richest sites are in temperate/arctic areas - fig 1 c), which seems to contradict the density vs richness argument. Would it be possible to see the raw richness data (as I understand it local species richness is calculated from Chao).

Response: Since our database is compiled based on many studies that differ in sampling effort and soil sample size, we believe that extrapolation of species richness is a necessary step for standardisation. To the best of our knowledge and data, the issue of singletons, even in tropical springtail communities, is not as severe as in aboveground tropical arthropod communities. To test if raw richness data would lead to different conclusions, we implemented additional linear mixed-effect modelling with “Total collection area” and “Sampled microhabitats” (only soil, only litter, and both) as additional predictors (LL 729-743). The results are presented in Extended Data Fig 10 and discussed in the main text (LL 309-315 and 338-340). The drivers of raw species richness are similar to that of extrapolated species richness, and the correlations with density and metabolism were also weak (all R² values were between 0.02 and 0.07; LL 249-250).

Re-response: I really appreciate the additional analyses and am happy that extrapolation is not causing any undue biases in the results and inference. I really like the side comparison between raw and extrapolated richness in ED Fig. 10. Thanks.

Response: You are welcome. Indeed, this was a very informative analysis that added robustness to our conclusions and also provided more context for our discussion.

Previous comment: Maybe a map of effort is needed, number of sites per- region/cell? And then one on the number of samples (where known).

Response: Agreed. We provide information on the sampling efforts in terms of the sampling-site distribution in Fig. 1a, methods, and Extended Data Figs 1, 2 and 3. Information on the number of samples is also displayed in Fig. 1a. We produced additional density-per-cell maps for the number of sites and samples which are now presented in Extended Data Fig. 2.

Re-response: Thanks for doing this. I really like this plot. I'd ask for one minor change and that is to report the results on the natural log scale. This will help reduce the influence of the few cells with very high sites/samples and hopefully identify any pattern (or none).

Response: Agreed. We now presented sampling density plots on a logarithmic scale.

Previous comment: I think the role of seasonality/phenology should be explored a little more in the model fitting methods. The authors claim that they have dealt with this issue, but it seems to me the best thing to do would be to include all available data and have a seasonal effect in the models (rather than choosing samples at their seasonal density/diversity peak). Or include a covariate that represents seasonal climate variability.

Response: We agree that seasonality/phenology is an important aspect affecting community parameters in some regions. It is a poorly-explored and difficult issue in the global modelling of soil biodiversity. We had to focus on the vegetation biomass peak periods globally, because this is what our data represent. Covariates which are related to climate variability (temperature seasonality, temperature annual range, precipitation seasonality, precipitation of the driest quarter) were included in both random forest and structural equation modelling (LL 633-639). However, they were not selected for the final structural equation model due to collinearity with other predictors. To further explore the potential seasonality bias, we now built additional linear mixed-effects models using only those sampling events where the sampling year and month were known (69% of all sites). We linked these data to specific year- and month-based data on air temperature and precipitation (CHELSA) and used 'Site' as random effect to account for interdependence of the events coming from the same site. Monthly temperature was analysed as offset from the annual mean for the given 7 site to avoid collinearity with the latter (LL 729-743). Results of this modelling showed that seasonal effects of climate are generally smaller than the effects of annual climatic means (see Extended Data Fig 10). However, while analysing sampling events, we discovered very strong effects of mean annual and monthly temperatures on community metabolism. This

implies high variation of this community metric with temperature. We believe this is an interesting avenue for future research and describe this in the main text (LL 309-315).

Re-response: Great! I really appreciate this additional analysis, very interesting about the potential effect of climate variability on the community metabolism. It would be fascinating to explore that relationship under future climate scenarios. There seems to be a mild (confidence intervals not overlapping zero) positive effect of monthly temperature on richness/density, which might make sense if there is higher richness/density in places that have lower mean annual temperature [MAT] regions (and typically have greater seasonal variability - temperate vs tropical regions). Which can be seen in ED Fig 8 and 10. It also looks like the relationship between MAT and raw richness/extrapolated richness is negative when using a linear term (ED Fig. 10). This would suggest higher richness in colder environments and potentially contradict the latitudinal gradients patterns. I personally think this is just a case that a unimodal (quadratic form) would likely do a better job at explaining this relationship (as is shown in Fig. 2) than a linear form (ED Fig. 10); it might be worth revisiting this model with polynomials for each covariate.

Response: We agree that it would be very interesting to explore metabolic effects under different climate scenarios in the future. We also re-built all linear models with quadratic relationship for the following factors: MAP, MAT, pH, SOC, %Clay, Elevation. These models, except one for metabolism, had a lower AIC than the linear ones, suggesting a better fit. We then tested if adding quadratic term for MAT only would improve the fit, which was the case specifically for extrapolated and raw species richness. Unfortunately, we did not find a way to display the results in an intuitive way that is comparable with the (linear) SEM model in the main text. Therefore, we added this information in the discussion in the main text, leaving the Figure unchanged.

Previous comment: I'd also think some type of sensitivity analysis needs to be done on the density/metabolism metrics, as these are derived from the data (not observed) choices in model coefficients (fits) could have a dramatic influence on the ratio between density/richness.

Response: Indeed, indirect estimations of the biomass and metabolism are sensitive to the used coefficients. To estimate the potential bias, we now used reported standard errors of coefficients in the length-mass and mass-metabolic rate allometric equations to calculate 'maximum' (mean coefficient + standard error) and 'minimum' (mean coefficient - standard error) biomass and community metabolism for each site (LL 548-552). We then run again the global extrapolations to include these uncertainties in our global estimates. The results are given in the text (LL 359-362): our global biomass estimates averaged 27.5 Mt C with 16.2 as the minimum and 28.8 Mt C as the maximum; our global community metabolism estimates averaged 15.2 Mt C month⁻¹ with 14.6 as the minimum and 18.6 Mt C month⁻¹ as the maximum

Re-response: This is a great addition, I think the authors for taking the time to redo these analyses and alleviate my previous concerns.

Response: You are welcome. We appreciate this suggestion and this additional analysis has improved robustness of our results.

Previous comment: Line 225, Yes, mean richness is higher in low latitudes, but looking at the raw data, there are clearly sites in Europe that are more species rich. Is there a reason for this? Like a difference in the sites over the less species rich ones (which there are plenty). I guess the point I'm getting to is there are random effects which can account for this difference? I.e. Agricultural vs forest? Or is this just a sampling artefact? More samples/more species?

Response: We believe that this is explained mainly by the sampling density. European temperate ecosystems are widely studied, because historically the taxonomic expertise concentrated in Europe. So, existing surveys observed both species-rich and species-poor communities. The pattern is unlikely to be explained by the habitat. For example, the global gradient for 'woodlands' (shown below) resembles the total gradient. We discuss this issue in the LL 350-358.

Re-response: Thanks for this explanation, and the discussion in the text.

Response: You are welcome. Thank you for emphasizing this topic.

Previous comment: For the richness analysis I wonder how much the rarefaction method is inflating/deflating richness in predictions. Looking at the maps of raw richness there doesn't seem to be the same major trend (mind you the colour scale is a bit hard to tell). I'm wondering instead of fitting a rarefied richness if the authors could fit a Poisson/negative binomial model with an offset for effort (number of sites per sample). Something like this is R should do the trick: `glm(richness ~ covariates + offset(log(No.Samples)),family=poisson)`

Response: We ran additional generalised linear mixed-effect models to predict raw species richness with "Total collection area" and "Sampled microhabitats" (only soil, only litter, and both) as additional predictors (LL 729-743). The results are now presented in Extended Data Fig 10 and discussed in the text (LL 355-358). The drivers of raw species richness are similar to those of extrapolated species richness

Re-response: Great, I appreciate the addition of raw species richness in the analyses. I was point towards the idea of tampering richness by effort (eg. using an offset). Rarefaction is trying to do this to some extent. You could think about in terms of say fisheries data, where they often use catch-per-unit-effort. This will give you the "corrected" count/biomass/weight of a species/community based on a relationship between effort (how many trawls/samples) and catch (what you get in your samples). There are a bunch of other issues (like does this relationship scale linearly, is effort biased towards higher numbers of the species), but it typically helps reduce bias in raw numbers and gives a better perspective of species numbers in relation to the effort used to collect/fish the target species. I'd like to make it clear that I

don't expect the authors to do any additional analyses here, but I was just pointing out how that might be useful for future analyses.

Response: We appreciate these additional explanations.